# Land Cover Change in the Blue Nile River Headwaters: Farmers' Perceptions, Pressures, and Satellite-Based Mapping

**Alelgn Ewunetu** [1,*]**, Belay Simane** [2]**, Ermias Teferi** [2,3] and **Benjamin F. Zaitchik** [4]

1 Department of Geography and Environmental Studies, Woldia University, P.O. Box 400 Woldia, Ethiopia
2 Center for Environment and Development Studies, Addis Ababa University, P.O. Box 3880 Addis Ababa, Ethiopia; belay.simane@aau.edu.et (B.S.); ermias.teferi@aau.edu.et (E.T.)
3 Water and Land Resource Center, Addis Ababa University, P.O. Box 3880 Addis Ababa, Ethiopia
4 Department of Earth and Planetary Sciences, Johns Hopkins University, Baltimore, MD 21218, USA; zaitchik@jhu.edu
* Correspondence: ewunetu.alelign@gmail.com; Tel.: +251-912-772-560

**Abstract:** The headwaters of the Blue Nile River in Ethiopia contain fragile mountain ecosystems and are highly susceptible to land degradation that impacts water quality and flow dynamics in a major transboundary river system. This study evaluates the status of land use/cover (LULC) change and key drivers of change over the past 31 years through a combination of satellite remote sensing and surveying of the local understanding of LULC patterns and drivers. Seven major LULC types (forest land, plantation forest, grazing land, agriculture land, bush and shrub land, bare land, and water bodies) from Landsat images of 1986, 1994, 2007, and 2017 were mapped. Agriculture and plantation forest land use/cover types increased by 21.4% and 368.8%, respectively, while other land use/cover types showed a decreasing trend: water body by 50.0%, bare land by 7.9%, grassland by 41.7%, forest by 28.9%, and bush and shrubland by 38.4%. Overall, 34.6% of the landscape experienced at least one LULC transition over the past 31 years, with 15.3% representing the net change and 19.3% representing the swap change. The percentage change in plantation forest land increased with an increasing altitude and slope gradient during the study period. The mapped LULC changes are consistent with the pressures reported by local residents. They are also consistent with root causes that include population growth, land tenure and common property rights, persistent poverty, weak enforcement of rules and low levels of extension services, a lack of public awareness, and poor infrastructure. Hence, the drivers for LULC should be controlled, and sustainable resources use is required; otherwise, these resources will soon be lost and will no longer be able to play their role in socioeconomic development and environmental sustainability.

**Keywords:** LULC change; drivers; pressures; North Gojjam sub-basin; remote sensing; GIS

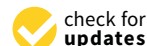

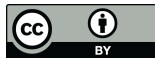

## 1. Introduction

Land use/cover (LULC) change is a persistent environmental issue encountered on a global scale [1,2], and Africa is experiencing significant changes across the continent. In recent decades, African grassland, woodland, and other vegetated areas have been increasingly converted into agricultural and settlement areas [2]. Africa lost 16% of its forests and 5% of its woodlands and grasslands during the period from 1975 to 2000, and more than 50,000 km$^2$ per year of natural vegetation has disappeared [3]. According to the same source, the majority of vegetation cover has been converted into agricultural and settlement land. LULC change in Ethiopia has followed a similar trend, with significant implications for land degradation and loss of ecosystem services. According to a global land degradation assessment, more than 26% of Ethiopian land has been degraded, and 30% of the population's livelihood has been affected from the period of 1981–2003 attributable to land degradation [4]. In Ethiopia, deforestation is an ongoing process, and it is acause of biodiversity loss, changing climatic conditions, desertification, and soil erosion [5–7]. Forest

cover in Ethiopia, which was more than 40% at the beginning of the 20th century, reduced to 2.36% in 2000 as a result of population growth [7], though the standardized LULC calculation method [8] is required to quantify the recent forest cover. The population size of the country has more than doubled in the last three decades from 40 million in 1984 to over 90 million in 2014, and it is estimated to reach 130 million by the year 2030 [9]. More than 85% of the population live in rural areas and highly depend on natural resources to sustain their way of life, and this results in an increasing resource demand in the country [9,10].

The Blue Nile (Abbay) basin, located in the western highlands of Ethiopia, is located in a critically important agricultural region. As it is also the location of the headwaters of the largest tributary of the Nile River, land and water dynamics in the basin have received particular attention from the research community. Several studies have been conducted on the biophysical state of change in the basin using remote sensing and GIS technologies. For example, Bewket [6] conducted an LULC study on the Chemoga watershed for the period of 1957–1998 and found that forest cover was increased. Teferi et al. [11] studied the Jedeb catchment for the period of 1957–2009 and confirmed that plantation forest and cultivated land cover increased at the expense of other land uses, but expansion of cultivated land ceased from the period of 1994–2009. Gebrehiot et al. [12] found that cultivated land increased alarmingly in place of forest land in Birr and Upper-Didesa for the period of 1957–2000. Bewket and Abebe [13] observed that the area of forest and dense tree covers decreased from the period of 1950–2001 in the Gish Abay watershed. However, these studies did not have sub-basin focus and did not include land users' knowledge, and the findings were contradictory, perhaps because of the study areas' heterogeneity in natural resource distribution and management systems [14,15]. Most of the previous studies also missed considerable biophysical factors, such as elevation and slope gradient in relation to individual land use types change. Furthermore, drivers of LULC change are dynamic, and they differ across regions depending on the dominant socioeconomic and biophysical factors. For example, Gessesse and Bewket [16] confirmed that the expansion of infrastructure at the expense of other LULC units in the Modjo watershed since the 1970sin the central highlands of Ethiopia was one of the drivers of landscape change. Expansion of farmland was confirmed by Gebrehiwot et al. [12], but not in the Jedeb watershed as tested by Teferi et al. [11] in the period of 1994–2009.In the Wollo area, population growth has been reported as a contributing factor for bush and shrub land expansion [17], but it has had a negative impact on bush and shrubland expansion in the Chemoga [6] and Jedeb watersheds [11]. Thus, it is worth noting that the causes of LULC change are also time- and location-specific. A driver identified a decade ago may not be valid in current times if interventions are made regarding the driving factor [18]. For example, deforestation was previously a key direct driver of land use change in the Wollo area of the Ethiopian highlands, but recently, reforestation has been identified as a dominant driver of land degradation [17]. Therefore, the biophysical environment is subjected to change caused by human activities, which have been continuously modifying ecosystem services [2,18,19]. Such continuous human intervention can also be observed in the upper Blue Nile basin of Ethiopia [6,14,20,21]. Though land degradation is a problem, efforts are being made by the government NGOs and the local community to engage in sustainable land management practices in the Abbay basin [14,21]. However, the recent land management effect on ecosystem rehabilitation is not well-studied, particularly in the North Gojjam sub-basin.

Remote sensing offers a powerful tool for LULC change analysis, but studies of LULC change based solely on remote sensing may not be relevant or trustworthy for locally specific environmental application. Integrated, place-based research on LULC change requires a combination of agent-based systems and narrative perspectives for an in-depth understanding of biophysical states [18,22,23]. The integration of remote sensing information with local land users' information can yield deeper insights into LULC change and the drivers of change. As a result, there is a growing need for the integration of scientifically proven knowledge with farmers' local knowledge of the state of land resources evaluation [18,22–24]. The farmers' knowledge was accumulated from day-to-day observations of

changes in the capacity of the ecosystem services to support their livelihoods. On the other hand, Landsat images were selected for years that align with major events that occurred in the study area. Accordingly, the 1986 image is indicative of conditions toward the end of the Dergue regime, following a period of collectivization of land resources in Ethiopia [11,25]. The year 1994 represents the period in the aftermath of the fall of the Dergue regime and the early years of rule by the Ethiopian People's Revolutionary Democratic Front (EPRDF). During this time, there was massive deforestation and expansion of agricultural land [26]. The 2007 image was selected to evaluate the introduction of sustainable land management programs in the headwater of Abbay [27]. Finally, to include recent changes and the current biophysical status in the study area, the 2017 Landsat image was selected. Therefore, this study evaluates the status of LULC change and key drivers of change over the past 31 years for a sub-basin at the source of the Blue Nile river (North Gojjam sub-basin) through a combination of satellite remote sensing and surveying of local understanding of LULC patterns and drivers. The specific objectives were (1) to quantify the extent, trend, and annual rate of LULC change in the period of from 1986 to 2017; (2) to obtain local people's perspectives on LULC change; and (3) to identify key biophysical and socioeconomic drivers of LULC change in the sub-basin.

## 2. Materials and Methods

### 2.1. Description of the Study Area

The North Gojjam sub-basin is located between 38.2° E to 39.6° E longitude and 10.8° N to 11.9° N latitude. It is one of the major tributaries of the Blue Nile/Abbay river (Figure 1). Its area covers about 1,431,360 ha, which stretches between the Choke and Guna mountains. The total population of the sub-basin and surrounding villages is 3,565,892 [15,21], with people settled in scattered and dispersed areas. The altitude of the sub-basin ranges from 1044 to 4048 masl (Figure 1). The dominant agroecological zone is characterized by tepid to cool moist middle highlands, and cold to very cold moist subafroalpine to afroalpine highlands. However, the eastern and south eastern parts of the sub-basin are hot to warm moist lowlands [15,21]. According to the Ethiopian National Metrological Agency (EMA) [28], the average maximum and minimum temperature of the sub-basin varies from 24.6–28.1 °C and 11.0–14.51 °C, respectively, and, generally, the mean annual temperature is 19.41 °C. The rainfall pattern is closely correlated with the annual migration of the intertropical convergence zone (ITCZ), and most of the rainfall occurs in summer, from June to September, with the highest amount in August [15,21]. The distribution of rainfall across the sub-basin is uneven; the highlands tend to be wetter than the lowlands. The meteorological record data from the stations within the sub-basin and the surrounding area (for the period of 1986–2017) indicated that the mean total rainfall is 1334.48 mm with a minimum of 810 mm and a maximum of 1815 mm [28]. Generally, the dominant climate condition of the sub-basin is the tropical highland monsoon [15,21]. According to these authors, the climate in the area is mainly influenced by altitude and global weather systems.

The dominant soil types are Leptosols, Vertisols, Luvisols, and Alisols. The geology of the sub-basin is mainly dominated by basalt, but the lowlands are dominated by sandstone [29]. Natural forest cover is low and found on riverbanks, hillsides, and churches to some extent. *Eucalyptus globulus* forest is dominant among the introduced trees, particularly in the highlands where its cover has shown an increase.

Unreliable rain-fed agriculture is the primary source of livelihood for the majority of the population in the sub-basin. It is characterized by a smallholder mixed crop and livestock production system. Different types of crops have been produced to a great extent, but vegetation and fruits have only been produced to a limited extent. The dominant crop types include cereals (wheat, barley, teff, maize, and sorghum), pulses (bean and pea), and oilseeds (niger seed, flux, and cabbage). The dominant livestock types are cattle, sheep, goats, horses, donkeys, mules, and poultry. Soil erosion, climate variability, and water

shortage explain the prevailing food insecurity and poverty in the sub-basin to a large extent.

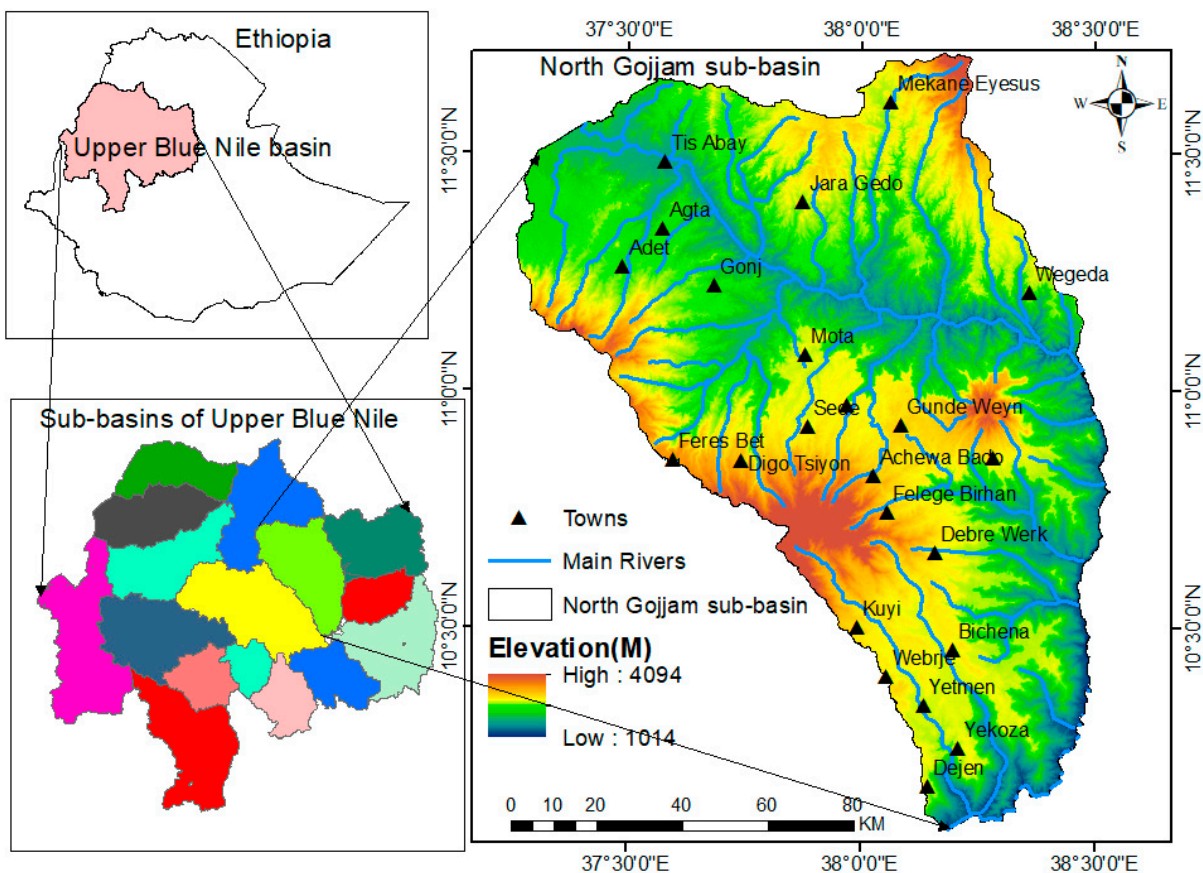

**Figure 1.** Location map of the North Gojjam sub-basin and its topography.

*2.2. Data Sources and Methodology*

2.2.1. Socioeconomic Data

While there are various methods to conduct socioeconomic surveys based on the aim of the study, focus group discussions (FGDs), in-depth interviews, and field observation were used to gain detailed information from local communities regarding the trends of LULC change and the importance of driving forces for changing landscape in the North Gojjam sub-basin between the years 1986 and 2017. To obtain in-depth and diverse information, FGD participants were selected from different agroecology and social classes based on their age, gender, and local knowledge (Table 1). There is no conclusive design regarding the number of participants in one FGD meeting, but a range of 6–10 participants is generally recommended. If the number of participants is lower than 6, the diversity of opinions to be offered may be restricted, while if there are more than 10 participants, it may be difficult for everyone to express their opinions in detail [30]. Thus, groups were limited to 6–10 members in this study. FGDs were facilitated by the corresponding author guided by a checklist of open-ended questions from May to June 2018. The questions covered the extent, trend, and the drivers of land use/cover change in the past and present situation in the locality. Moreover, questions covered challenges to cease and reduce the drivers of ecosystem change. In each of the studied villages, two series of FGDs were held at development agents' (DAs') offices and churches. A total of 18 FGDs were carried out in 9 villages with 127 (78 male and 47 female) community members selected from upper, middle, and lower parts of the sub-basin (Table 2). In-depth interviews were also held with 27 farmers, 9 DAs, and 9 districts' crop, livestock, and natural resource experts to

obtain additional information regarding the study area. All FGD participants were not included in the in-depth interview. The interview was held by the first researcher using an open-ended questionnaire at the DA offices, churches, and farm fields. Using this method, detail information regarding landscape change and the drivers of change regarding the period of the last 31 years (1986–2017) was collected. To gain a better understanding of the major observed problems of the sub-basin, transect walks and informal talks with farmers were conducted.

**Table 1.** Distribution of sample respondents.

| Zone | District | Village | Agroecology | Total HH |
|:---:|:---:|:---:|:---:|:---:|
| East Gojjam | Enarj Enauga | Koso-zira | Upper | 12 |
| | | Titar Badima Yizar | Middle | 18 |
| | | Gedeb Georgis | Lower | 13 |
| West Gojjam | Dega Damot | Z/Wogem | Upper | 17 |
| | | A/Medhanyalem | Middle | 14 |
| | | G/T/Haymanot | Lower | 14 |
| South Gonder | Andabet | Gota | Upper | 13 |
| | | Yedidi Gimegne | Middle | 14 |
| | | Genete Mariyam | Lower | 14 |
| Total | - | - | - | 127 |

### 2.2.2. Spatial Data

Four Landsat images were downloaded from the US Geological Survey (USGS) Earth Explorer (http://earthexplorer.usgs.gov) [31] for land LULC classification analysis. On the other hand, ASTER-GDEM was obtained from the Aster Global Digital Elevation Model version 3 (http://gdex.cr.usgs.gov/gdex/) [32]. All Landsat images were obtained from the month of January, the dry season, to avoid major differences in vegetation phenology and to gain less cloud cover images [33–35]. We used comparable bands for LULC classification: band 1–7 for Landsat thematic mapper (TM) and enhanced thematic mapper (ETM)+ (with the exclusion of band 6) and band 2–7 for Landsat operational land imager/thermal infrared sensor (OLI/TIRS) (Table 2). TM and ETM are the two sensors in Landsat, and they have been in use since 1982. The sensors are important for LULCC analyses. Obtaining adequate datasets requires the selection of type of sensor, relevant wavelength bands, and date(s) of acquisition [36]. Landsat 8(OLI/TIRS) was launched in 2013 by improving the past landsite qualities, and it was applied for the observation of land use and cover for this period [31].

### 2.2.3. Ground Truth Data

Ground truth data for LULC calcification and accuracy assessment were collected using different information. The reference data for the 1986 and 1994 images were collected based on interviewing local elders about known locations in the locality. These elders were not involved in both FGDs and the in-depth interview in this study. Information collected from elderly people in a participatory approach is a good source of reference data for validation of old LULC maps. Old toposheets were used to initiate the discussion among the group members of the elderly people, while the 2007 image reference data were collected from Google Earth using a time slider image [37]. For the recent image (2017), reference data were collected based on field observations using a handheld global positioning system (GPS) instrument for recording reference truth points (latitude and longitude). Using a simple random sampling technique, 2500 sample points were collected from the representatives of LULC classes for each study year. Of these total ground truth points, 547, 587, 724, and 738 were used for accuracy assessment of image classification for the years 1986, 1994, 2007, and 2017, respectively.

**Table 2.** List of time series Landsat data used for the study.

| Sensor | Path/Row | Acquisition Date | Resolution | Source |
|---|---|---|---|---|
| 1986 TM | 170/052 | 19 January 1986 | $30 \times 30$ | USGS |
| | 169/052 and 053 | 28 January 1986 | $30 \times 30$ | USGS |
| 1994 TM | 170/052 | 25 January 1994 | $30 \times 30$ | USGS |
| | 169/052 and 053 | 18 January 1994 | $30 \times 30$ | USGS |
| 2007 ETM+ | 170/052 | 21 January 2007 | $30 \times 30$ | USGS |
| | 169/052 and 053 | 17 January 2007 | $30 \times 30$ | USGS |
| 2017 OLI | 170/052 | 24 January 2017 | $30 \times 30$ | USGS |
| | 169 /052 and 053 | 17 January 2017 | $30 \times 30$ | USGS |

2.2.4. Satellite Image Preprocessing

Satellite image preprocessing operations were carried out using ArcGIS 10.5 [38]. Because of the technical conditions of the sensors and platforms, and the state of the atmosphere, earth rotation and terrain effects, the images acquired by the sensors can be subject to errors and distortions [39,40]. To reduce these effects and to facilitate effective monitoring of biophysical change, image preprocessing is vital [41]. Image preprocessing can be achieved through various radiometric and geometric correction techniques [42]. Such radiometric techniques are a prerequisite for creating good quality sensor data and a higher level of downstream products [42]. This helps to reduce the variability among scenes and increases the comparability of data obtained from a different time and different sensors [42]. Landsat satellite sensors' digital number (DN) values were converted to radiance and then to top-of-atmosphere (TOA) reflectance [41]. The images acquired from the Earth Explorer website are in this format (https://earthexplorer.usgs.gov/). Radiometric calibration of the, TM, ETM+, and OLI sensors involves rescaling of the raw digital numbers (Q) transmitted from the satellite to calibrated digital numbers ($Q_{cal}$), which have the same radiometric scaling for all scenes processed on the ground for a specific period. Thus, we used the following equation previously used by Chander et al. [42] to perform the $Q_{cal}$-to-spectral radiance conversion using QGIS3.1 [43].

$$L_\lambda \ = \ (\frac{LMAX_\lambda - LMIN_\lambda}{Q_{calmax} - Q_{calmin}}) \, (Q_{cal} - Q_{calmin}) \ + \ LMIN_\lambda \tag{1}$$

where

$L_\lambda$ = spectral radiance at the sensor's aperture (W/(m$^2$sr μm));
$Q_{cal}$ = quantized calibrated pixel value (DN);
$Q_{calmin}$ = minimum quantized calibrated pixel value corresponding to $LMIN_\lambda$;
$Q_{calmax}$ = maximum quantized calibrated pixel value corresponding to $LMIN_\lambda$;
$LMIN_\lambda$ = spectral at-sensor radiance that is scaled to $Q_{calmin}$ (W/(m$^2$ sr μm));
$LMAX_\lambda$ = spectral at-sensor radiance that is scaled to $Q_{calmax}$ (W/(m$^2$ sr μm)).

The second step of the radiometric calibration operation was the conversion of the at-sensor spectral radiance to top-of-atmosphere (TOA) reflectance. This has three advantages: (1) it removes the cosine effect of different solar zenith angles due to the time difference between data acquisitions; (2) TOA reflectance compensates for different values of exoatmospheric solar irradiance arising from spectral band differences; (3) the TOA reflectance is corrected to the variation in the Earth–Sun distance between different data acquisition dates [44]. TOA reflectance was computed using Equation (2) in QGIS3.1 [43].

$$P_\lambda \ = \ \left( \frac{\pi \, * \, L_\lambda \, * \, d^2}{ESUN_\lambda * \, \cos\theta_S} \right) \tag{2}$$

$P_\lambda$ = planetary TOA reflectance (unitless); $\pi$ = constant with an approximate value of 3.14159 (unitless); $L_\lambda$ = spectral radiance at the sensor's aperture (W/(m$^2$srμm)); $d$ = Earth–Sun distance (astronomical units); and $ESUN_\lambda$ = mean exoatmospheric solar irradiance

(W/(m²sr μm)). The parameters used for Landsat TM and ETM+ images for radiometric calibration are available in the metadata in the text file format. The value of exoatmospheric solar irradiance was summarized for the TM and ETM+ sensors using the Thuillier solar spectrum [45]. For Landsat 8 (OLI 2017), the digital numbers (DNs) of the images were directly converted to TOA reflectance using the formula given in the U.S. Geological Survey [46] using QGIS3.1 software [43].

$$L_\lambda = \frac{M_L\,Q_{col} + A_L}{\sin\theta} \tag{3}$$

where $L_\lambda$ = spectral radiance (W/(m²srμm)); $M_L$ = radiance multiplicative scaling factor for the band (RADIANCE_MULT_BAND_n from the metadata); $A_L$ = radiance additive scaling factor for the band (RADIANCE_ADD_BAND_n from the metadata); $Q_{col}$ = L1 pixel value in DN; and $\theta_S$ = solar elevation angle. The focal analysis or gap filling was completed to build bad lines in Landsat ETM+7 before radiometric correction using ERDAS IMAGINE14 [47]. However, Landsat images acquired from USGS were already georeferenced into WGS_84_UTM_ zone_37N and datum_D_WGS_1984, and they did not fit together. For instance, the 2017 Landsat imaged does not perfectly fit with other images. For this reason, we first georeferenced the 2017 images using the georeferenced topographic map. Then, other images were registered via the image-to-image georeferencing technique using the 2017 images as base images in ERDAS IMGINE [47]. Satisfactory ground control points (GCPs) were taken from permanent points, such as the river and stream intersection. Each image's root mean square error (RMSE) result was less than 0.29 pixels. Then, the images were reprojected to the same projection system (WGS_84_UTM_zone_37N and datum_ D_WGS_1984) and resampled to 30 m resolution via the nearest neighbor algorithm. There are various Landsat image enhancement techniques to aid visual interpretation and clear visual appearance of the image. However, histogram equalization was applied in this study, as this method is the best technique for the frequency distribution of the large pixel values through the image. After subsequent preprocessing, all images were in mosaic form, and the sub-basin study images were masked using a digitizing shapefile of the sub-basin with ArcGIS10.5 tools [38].

### 2.2.5. Image Classification

In this study, unsupervised classification was initially applied prior to the field survey using the visual interpretation method to differentiate various land use/cover types in the studied sub-basin. Then, supervised image classification was applied after the collection of training data from existing LULC classes. Using ERDAS IMGINE (version 14) [47] from each of the predetermined LULC classes, signatures of polygon were delineated based on the information obtained from field observations, local people, Google Earth, and the image visual interpretation through false-color composite interpretation [48]. The use of high-resolution Google Earth images to derive the ground truth for land use classification accuracy has been suggested in previous studies [49,50]. Spectral signatures of the respective land use/cover class derived from the satellite imageries were recorded using the pixels enclosed by these polygons. A polygon of the homogenous map class was selected as a sample unit instead of a singlepixel to reduce misregistration and to increase the quality of match of image segmentation analysis. Landsat pixels that overlap the training areas were then used to perform the classification [51]. A stratified random sampling method was used to collect an optimum number of sample reference polygons for the classification. A satisfactory spectral signature ensures minimal confusion in the mapped land cover. The number of training sites varied from one LULC to another depending on ease of identification and level of variability [52]. Then, each pixel of the image dataset was placed in an LULC class via maximum likelihood classifier, and outputs were presented in different tables and figures. The maximum likelihood classifier (MLC) is the most widely adopted parametric classification algorithm, and it uses a per-pixel

method to account for the spectral information of LULC classes [11,25,52,53]. Generally, the steps of the current LULC change classification procedure is presented in Figure 2.

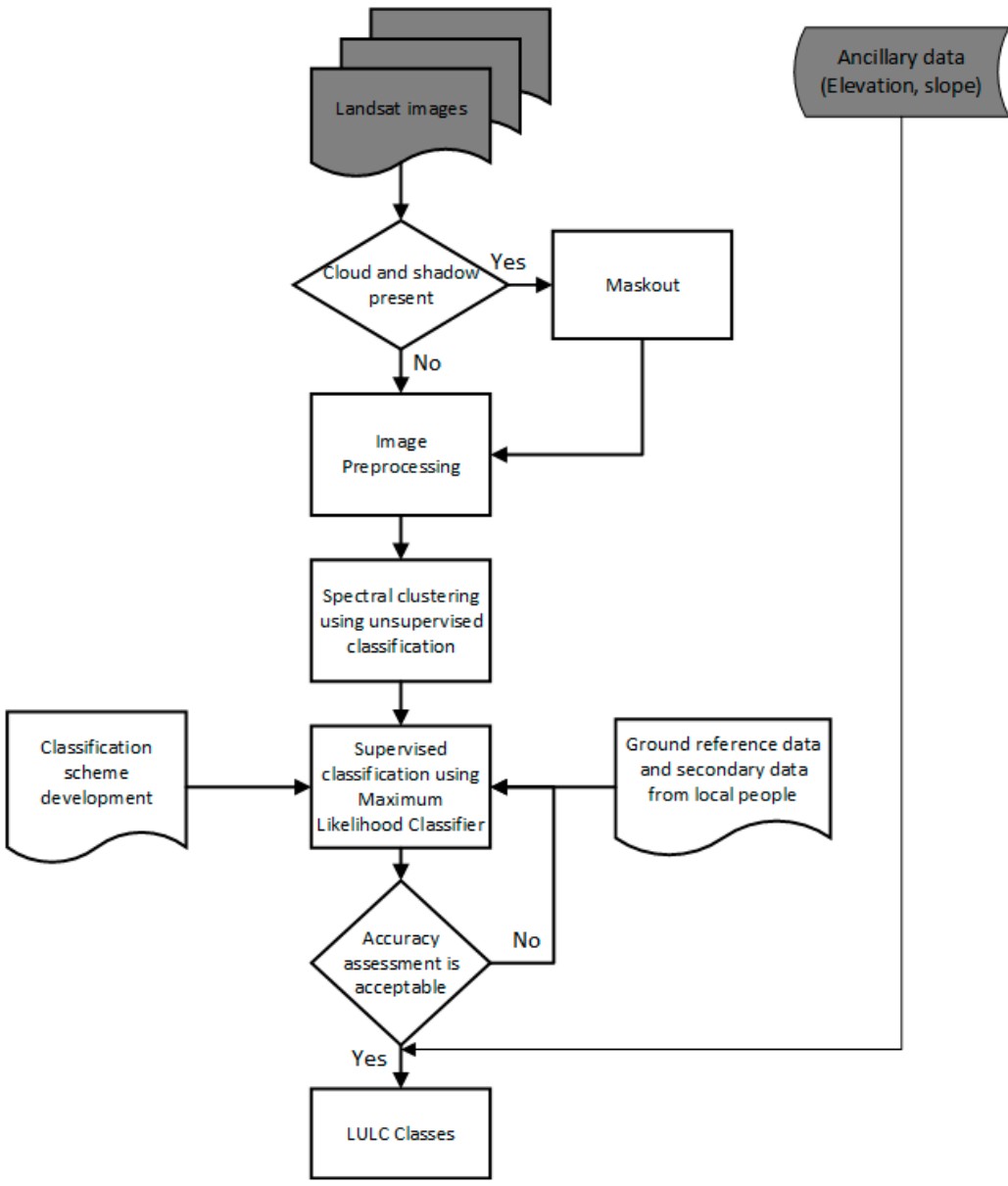

**Figure 2.** Flowchart showing the procedures used to arrive at the final land use/cover (LULC) map.

The nomenclature for LULC types was given based on the authors' prior knowledge about the study area, brief field survey, and Google Earth map observation [37]. All LULC types have a given clear and precise description as they have distinct differences from each other (Table 3). The classification of LULC types from a satellite image of the study area depended on the purpose, nature of the study area, and resolution of the satellite imagery. For example, we classified settlements as agriculture land for two reasons. The first reason is that during 1986, most houses' roofs were built from grass and straw, and they look considerably like cropland in the images. On the contrary, in 2017, most of the houses were built from tin and thus do not have similar reflectance as 1986 house types. Second, the study sub-basin is primarily rural; most houses are scattered and small in size, and they are almost completely surrounded by cropland. This makes it difficult to separate settlements from cropland at a 30-m resolution in Landsat images.

**Table 3.** Major land use/cover classes and their description.

| LULC Classes | Description |
|---|---|
| Agriculture land | The area covered with crop cultivation. This land use type includes rural settlements fenced with trees that are commonly found around homesteads and towns. This class also includes homesteads and the scattered trees on farmlands. |
| Water bodies | An area of land covered with surface water bodies such as lakes, rivers, and ponds. |
| Bare land | Areas under degraded lands and with some areas that are of bare ground, including sand, gravel, bedrocks, and riverbed gravels. |
| Grassland land | The area covered by permanent grass that is used for communal and private grazing lands. This class also includes rangelands. |
| Forest land | Areas covered by dense natural trees forming closed or nearly closed canopies, mainly growing naturally in the reserved land and along the riverbanks and the hillsides. |
| Plantation forest | Areas composed of transplanted seedlings of plants, mainly *Eucalyptus globulus*, junipers, and bamboo trees. |
| Bush and shrub | Land covered by bush and shrub land vegetation. This class also includes sparse trees on shrub and bush land. |

### 2.2.6. Derivation of Topographic Attributes

Topographic characteristics of altitude and slopes were generated from ASTER GDEM [38] using Arc GIS10.5 [37]. Applying the classification system developed by Bewket and Teferi [48], the slope class in the study area was reclassified into six classes: very gentle (0%–5%), gentle (5%–10%), moderate (10%–20%), steep (20%–30%), very steep (30%–50%), and extremely steep (50%–100%) slopes. Likewise, the elevation range was divided into six classes with an interval of 500 m elevation range. Finally, by overlaying the classified maps of each study year (1986 and 2017) on the elevation and slope maps, thematic information showing the relationship between LULC distribution and changes in each class of topographic variables was extracted using the ArcGIS10.5 tool [37].

### 2.2.7. Method of Socioeconomic Data Analysis

First, all 18 FGDs participants discussed, identified, and ranked the key drivers of land use/cover change based on their severity at all of the selected villages. Then, the results obtained from each group were added and changed to percentage total for aggregate results. Moreover, qualitative information gathered from key informant interviews, observation, and FGDs was used to triangulate, substitute, and complement quantitative data/Landsat information.

### 2.3. Postclassification Processing

#### 2.3.1. Classification Accuracy Assessment

To validate the quality of information derived from remotely sensed data, accuracy assessment is a key step in the process of remote sensing data analysis [52,54,55]. Accuracy assessment in spectral image classification is measured through a set of reference pixels via creating an error matrix [44,52]. Accordingly, all Landsat image classification accuracy levels of this study were checked using the error matrix rule, which allowed us to evaluate the kappa coefficient, overall accuracy, and the producer's and user's accuracy. Each supervised LULC classification validation was analyzed using more than 540 validation points.

2.3.2. Analysis of LULC Change Detection

Gross loss, gross gain, net change, and swap of LULC change for each class were calculated using the transition matrix methodology used by Teferi et al. [11] and Zewdie and Csaplovies [34] by applying it to each pair of sequential images: 1986–1994, 1994–2007, 2007–2017, and the entire period,1986–2017. In this approach, the transition matrix contains rows that include land cover classes at time 1 and columns for land cover categories at time 2. The proportion of land that transitions from category *i* to category j between the two time periods is denoted by $P_{ij}$. Persistence appears along the diagonal of the matrix and is denoted by $P_{jj}$. The entries off the diagonal indicate a transition from category *i* to a different category, namely category j. According to Pontius et al. [56], the proportion of the landscape in category *i* in time 1 ($P_{i+}$) is the sum of ($P_{ij}$) over all *j*, and the proportion of the landscape in category *j* in time 2 ($P_{+j}$) is the sum of ($P_{ij}$) over all *i*. The gains were computed from the difference between column total and persistence, while the losses were computed from the difference between the row totals and persistence. The total change ($C_j$) of a land use category was computed using the sum of gains and losses. The total change includes swap change ($S_j$), which denotes simultaneous gain and loss of a category on the landscape. The amount of swaps of land class j was calculated as two times the minimum of the gain and loss to create a pair of grid cells that swap each grid cell that gains with a grid cell that loses [11]. Net changes ($D_j$) are the differences between gain and loss of the successive/use class.

$$S_j \; = \; 2 * MIN(P_{j+} - P_{jj}, P_{+j} - P_{jj}) \tag{4}$$

$$C_j \; = \; D_j + S_j \; = \; P_{j+} + P_{+j} - P_{jj} \tag{5}$$

$$D_j \; = \; |P_{+j} - P_{j+}| \tag{6}$$

The annual rate of LULC changes between study periods and was calculated using the standard formula derived from the compound interest law recently applied by Teferi et al. [11], as it is a standardized method of land use classification with good estimation and biological meaning classification [8].

$$R_\Delta \; = \; \left(\frac{1}{T}\right) \times ln\left(\frac{A_2}{A_1}\right) \times 100 \tag{7}$$

where $R_\Delta$= average annual rate of change (%); $A_1$ = amount of land cover in time 1; $A_2 A_2$ = amount of land cover in time 2; and $T$ = the time interval between the two study years.

## 3. Results

### 3.1. Local Views on Direct Drivers of LULC Change in the North Gojjam Sub-Basin

In Ethiopia, deforestation is an ongoing problem that causes ecosystem service change and fragmentation [6,10,57,58]. The impact of deforestation and vegetation clearance for LULC change was ranked by 70.9% of FGDs participants' in the North Gojjam sub-basin (Figure 3). According to the local viewpoint, natural vegetation covers have been declining for the last 31 years, mainly associated with agricultural land expansion. According to local inhabitants, wood cutting for domestic energy, charcoal production, building, wood selling, furniture making, and farm tools continue to be factors for the forest, bush, and shrubland degradation. As a result, natural vegetation has significantly diminished and is currently observed only in churches, riverbanks, and the hillside areas. However, farmers explained in interviews that plantation forests (*Eucalyptus globulus*) have been increasing in the sub-basin, particularly in temperate and subtemperate agroecology zones. Due to the low productivity of cropland and land degradation, farmers have been begun to use their land for commercial tree plantation, particularly in the highland regions. This results in replacing of other indigenous trees, farmland, bush and shrubs, and wetlands and grassland covers. However, there is a challenge regarding reforestation and afforestation of indigenous plants in the sub-basin due to the shortage of land, free grazing, shortage of

water, and lack of follow-up after planting. Hence, vegetation cover clearance was the key driver of land use change in the study area.

Livestock pressure and poor management of grazing land are direct drivers of LULC change [10,11]. Similar to this, farmers in the FGDs and in-depth interview reported that the number of low productive livestock is higher than before and has been increasing with the increasing population growth but decreasing per household. Regarding this, 21.3% and 65.3% of FGDs participants ranked livestock pressure in first and second place, respectively, as it is a direct driver of LULC change in the North Gojjam sub-basin (Figure 3). Likewise, according to local experts, livestock is not improved in genetic potential, and, thus, they are poor in productive capacity. The feeding system also remains undeveloped and mainly depends on free grazing and crop residue/straw. Recently, the livestock grazing system has been shifted to bush and shrub lands because of conversion of grazing land to agriculture land. This results in the emergence of key derivers of vegetation cover clearance and soil erosion in the sub-basin. Even if farmers are recommended by experts to reduce livestock numbers and to use cut-and-carry systems for feeding, most farmers are not willing to do so. Farmers reported that the cut-and-carry system is difficult for feeding sheep and goats and that it is also labor-intensive, particularly in the temperate area where most farmers' livelihoods depend on sheep and horses. Thus, it is a challenge to implement a cut-and-carry system, particularly among landless youths, as they depend on communal grazing land. Similarly, farmers perceived that feeding animal using a cut-and-scarry system makes animals weaker than those feeding via free grazing. The FGD and in-depth interview participants also indicated that farmers share manpower, livestock, and waterway tracks on land with a steep slope. This aggravates high soil erosion in the rainy season and causes the formation of gullies. In summary, overgrazing and poor grazing practices result in land use change and interfere with ecosystem rehabilitation efforts through afforestation and reforestation in the sub-basin.

Using biomass for fuel energy is one of the key direct causes of land degradation in Ethiopia [10,57,58]. According to FGD participants, almost all people in the sub-basin used firewood followed by cow dung and crop residue for cooking, heating, and lighting. Collecting firewood and charcoal production is recognized by 64.6% of participants as the third direct cause of LULC change in the studied sub-basin (Figure 3). According to them, not only rural people but also those living towns depend on charcoal and firewood energy for cooking and heating. Poor land management and inappropriate land use affect ecosystem services and can lead to failure in land productivity [59,60]. Similarly, in the study area, farmers in the informal discussions confirmed that poor agricultural activities, such as cultivating steep slope land, plowing up and down (criss-cross tillage), and poor grazing systems were key causes of LULC change and severe soil erosion in the sub-basin. This confirms the notion that the farming system is a proximate cause of land resource conversion in the sub-basin. However, FGD participants did not rank poor land management among the top five direct drivers of LULC change.

Natural factors are key drivers of land use change in the North Gojjam sub-basin. FGDs and in-depth interview participants noted erratic rainfall and rugged topography as biophysical factors contributing to LULC change in general, and to land degradation, particularly in the study area. According to them, the sub-basin is well known for its irregular relief features, which include mountains with steep slopes, hills, gorges, and plains. These terrain features provide running water with energy to wear away topsoil. Approximately 26% and 21% of participants ranked topography as the fourth and fifth most important factor among the drivers of land use change, respectively (Figure 3). The sub-basin has unimodal rainfall that occurs during the summer season, extending from June to September, which results in high soil erosion mainly in the early summer season. In addition, soil erosion generally causes the removal of fertile portions of topsoil; the creation of degraded land cover types; and the formation of gullies, bare lands, and rock levels in the study area.

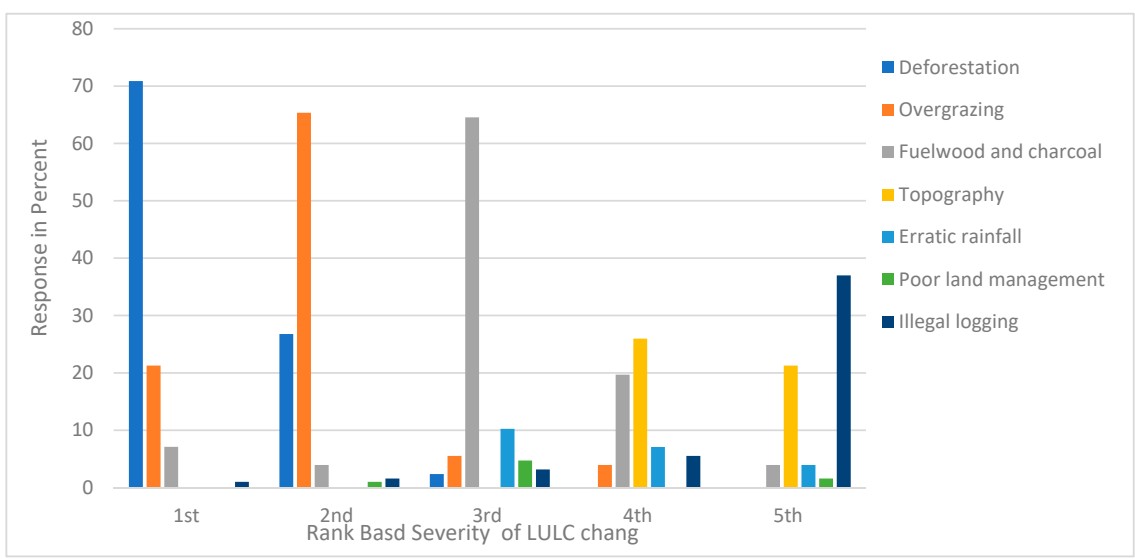

**Figure 3.** Direct drivers of LULC change in the North Gojjam sub-basin.

### 3.2. Landsat LULC Mapping

Having gathered local land users' perspectives on key LULC trends and the pressures that drive them, we now examine the representation of these changes in the satellite data.

### 3.2.1. Image Classification Accuracy Assessment

The accuracy assessment result of LULC for this study shows that for 1986, overall accuracy was 89.8% with a kappa coefficient of 0.9; for 1994, overall accuracy was 92.6% with kappa coefficients of 0.8. Similarly, in 2007, the overall accuracy was 87.8%, while the kappa coefficient was 0.8. In 2017, the overall accuracy was 91.6%, and the kappa coefficient was equal to 0.9. The user's accuracy of the individual LULC class ranged from 76.7% to 95%, and the producer's was found between 75.7% and 95% in all classification years (Table 4). Previous authors have set different thresholds for determining whether LULC classification is acceptable. Congalton [61] states that accuracy is "very good" if the overall classification accuracy result is above 81% [53], whereas the USGS guidelines call for overall accuracy is greater than 85% [62]. According to Ismail et al. [63], the agreed criteria for kappa (K) statistics are classified as poor if K < 0.4, good if 0.4 < K < 0.7, and excellent if k > 0.75. By all of these measures, the LULC classification maps generated in this study qualify as high-quality. Each LULC class validation was checked by more than 540 reference points.

**Table 4.** Accuracy of LULC classification of the North Gojjam sub-basin (1986–2017).

| LULC Classes | 1986 (%) | | 1994 (%) | | 2007 (%) | | 2017 (%) | |
|---|---|---|---|---|---|---|---|---|
| | Producer's | User's | Producer's | User's | Producer's | User's | Producer's | User's |
| CL | 90.4 | 92.5 | 89.6 | 94.9 | 88.9 | 90.7 | 96.8 | 92.8 |
| WB | 93.3 | 75.7 | 96.0 | 88.9 | 96.0 | 88.9 | 86.7 | 100.0 |
| BL | 90.2 | 93.9 | 83.3 | 95.3 | 76.7 | 95.8 | 90.1 | 87.6 |
| GL | 89.8 | 87.2 | 92.0 | 81.8 | 90.0 | 82.4 | 92.4 | 94.0 |
| FL | 86.0 | 90.5 | 90.4 | 90.4 | 78.6 | 90.2 | 88.4 | 87.5 |
| PL | 79.0 | 94.4 | 78.0 | 80.0 | 79.0 | 79.0 | 89.3 | 89.3 |
| BSL | 93.6 | 86.5 | 90.0 | 85.7 | 92.3 | 85.7 | 79.1 | 91.3 |
| Overall | 89.8 | | 92.6 | | 87.6 | | 91.6 | |
| Kappa | 0.9 | | 0.8 | | 0.8 | | 0.9 | |

Note: CL = cultivated land and settlement, WB = water body, FL = forest land, BSL = bush and shrubland GL = grazing land, PL = plantation, BL = bare land.

### 3.2.2. Analysis of Land Use/Land Cover Change in the North Gojjam Sub-Basin

The distribution of LULC types in the North Gojjam sub-basin for the years 1986, 1994, 2007, and 2017 are presented in Figures 4 and 5. The area extent, trend, annual rate of change, and matrix of land use and cover results of the sub-basin are provided in Tables 5–8.

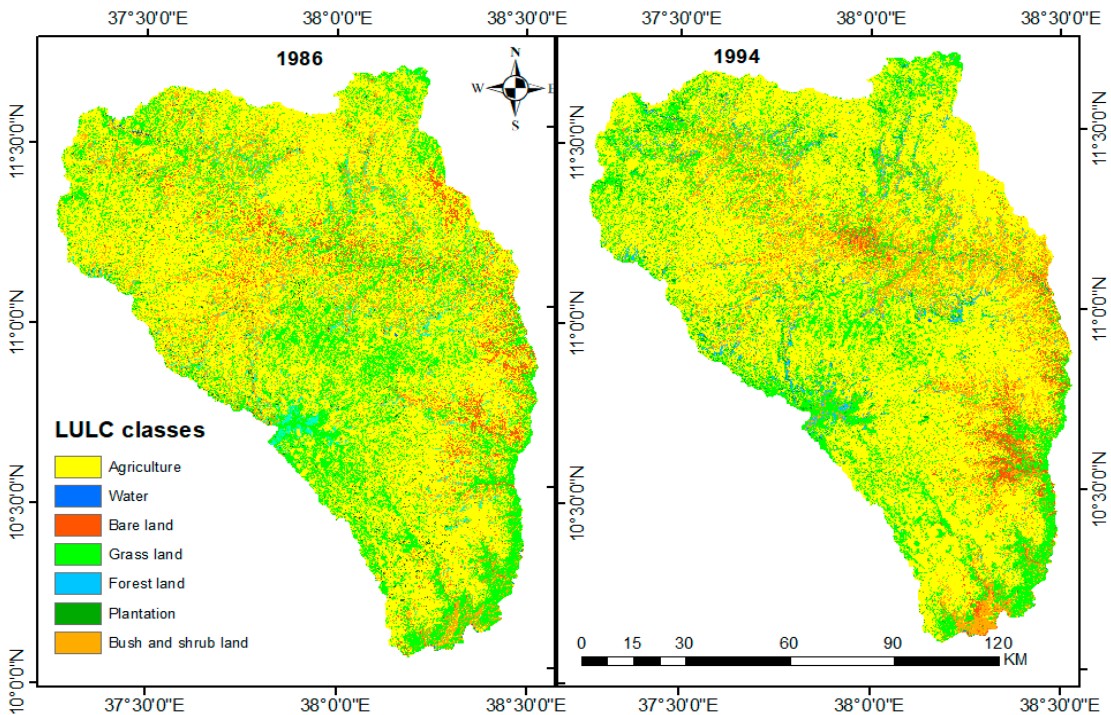

**Figure 4.** 1986 and 1994 LULC maps of the North Gojjam sub-basin.

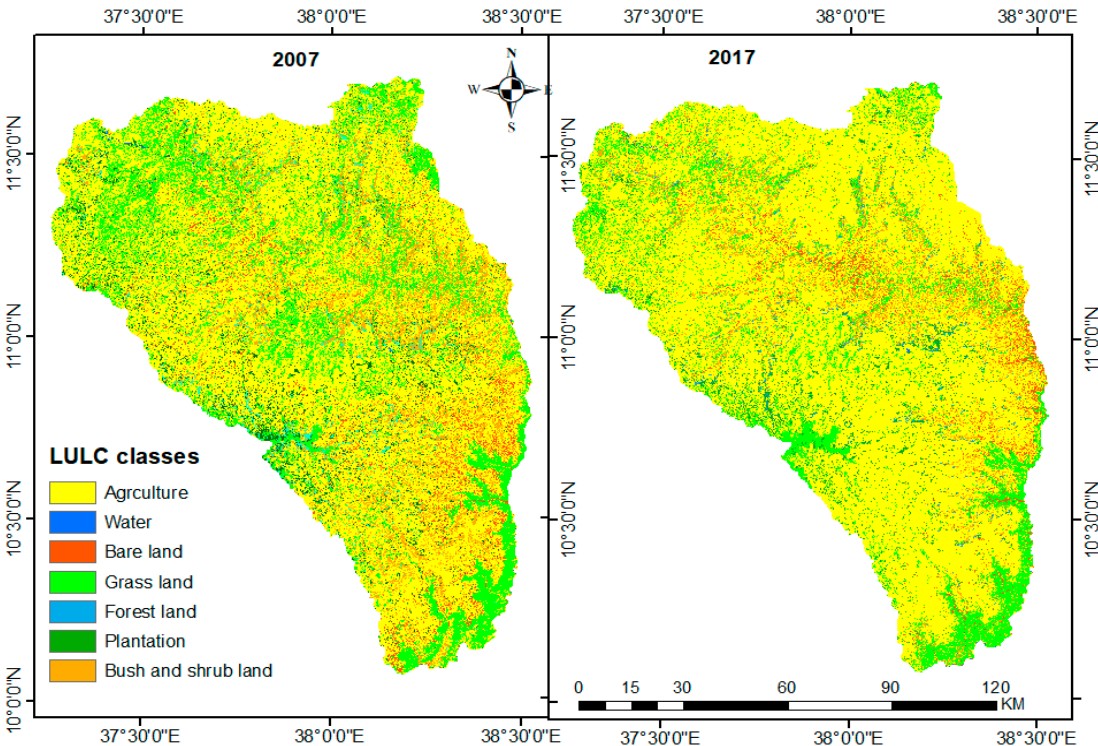

**Figure 5.** 2007 and 2017 LULC maps of the North Gojjam sub-basin.

As displayed in Tables 5 and 6, significant LULC change occurred over the last 31 years in the North Gojjam sub-basin. At the beginning of the study period (1986), agricultural land was the most dominant type, covering 58.2% of the total sub-basin area, followed by grazing land (21.4%), bush and shrub land (13.43%), bare land (3.2%) natural forest (2.9%), plantation forest (0.8%) and water bodies (0.08%). During the first study period time interval (1986–1994), agriculture cover increased from 58.2% to 61.7% in areal extent and plantation forest showed an increase of 105.2% from its original size. In this period, some degraded land was rehabilitated. Similarly, local land users confirmed that the afforestation and reclamation measures expanded due to mass plantation campaigns during this period on communal grazing land, bare land and bush and shrub land areas. Conversely, natural forest, bare land, bush and shrub land, and grazing land covers declined by 27.5%, 16.1%, 14.1% and 5.1% during the period of from 1986 to 1994, respectively (Table 6).

**Table 5.** LULC distribution of the North Gojjam sub-basin for 1986, 1994, 2007 and 2017.

| LULC | 1986 | | 1994 | | 2007 | | 2017 | |
|---|---|---|---|---|---|---|---|---|
| | Area (ha) | Area (%) | Area (ha) | Area (%) | Area (ha) | Area (%) | Area (ha) | Area (%) |
| CL | 833,337.79 | 58.22 | 883,292.26 | 61.71 | 952,283.8 | 66.53 | 1,011,542.1 | 70.67 |
| WB | 1145.09 | 0.08 | 1288.22 | 0.09 | 1001.95 | 0.07 | 572.54 | 0.04 |
| BL | 45,230.98 | 3.16 | 37,931.04 | 2.65 | 23,474.30 | 1.64 | 41,652.58 | 2.91 |
| GL | 306,740.45 | 21.43 | 290,995.4 | 20.33 | 215,992.2 | 15.09 | 178,776.86 | 12.49 |
| FL | 41,652.58 | 2.91 | 30,201.70 | 2.11 | 32,348.74 | 2.26 | 29,629.15 | 2.07 |
| PL | 11,021.47 | 0.77 | 22,615.49 | 1.58 | 47,807.42 | 3.34 | 51,672.10 | 3.61 |
| BSL | 192,231.65 | 13.43 | 165,178.9 | 11.54 | 158,451.5 | 11.07 | 117,514.66 | 8.21 |
| Total | 1,431,360 | 100 | 1,431,360 | 100 | 1,431,360 | 100 | 1,431,360 | 100 |

Note: CL = cultivated land and settlement, WB = waterbody, FL = forest land, BSL = bush and shrub land, GL = grassland, PL = plantation, BL = bare land.

In the second study period (1994–2007) agriculture land increases from 61.7 to 66.5% in areal extent as a result of cropland and settlement expansion into grazing land and natural vegetation (Table 5). During the same period, land covered by plantation and natural forest increased by 333.8% and 7.1%, respectively (Table 6). According to local farmers' view, during this period, bush and shrubland covers were regenerated into dense forest. In addition, afforestation and reforestation practices were applied to expand forest cover. These increases in agriculture, plantation, and forest land cover were at the expense of other land covers: bare land, grazing land, bush and shrubland, and water bodies decreased by 48%, 29.6%, 17.6%, and 12.5%, respectively (Table 6). The result indicates that, like in the first years, the trends of agriculture and plantation expansion continued.

In the third study period (2007–2017), the expansion of agriculture land and plantation continued, increasing from 66.7 to 70.7% and 3.3 to 3.6%, respectively (Table 5). In this period, the degraded land cover also increased, with an areal extent of change of 77.2%. However, unlike the previous period, forest cover slightly decreased from 2.3 to 2.1% in areal extent (Table 5). Similarly, waterbodies, bush and shrubland and grazing land were reduced by 42.9%, 25.8%, and 17.2%, respectively (Table 6). Generally, though specific trends varied, the pattern of landscape change showed a continued tendency toward more land being brought under cultivation, settlement and plantation at the expense of other LULC classes in the sub-basin. In the whole study period (1986–2017), although fluctuating rates and trends were observed, agriculture land showed a noteworthy increase of 58.2–70.7% at the expense of natural vegetation and grazing land (Table 5). Similarly, the magnitude of plantation forest cover showed a remarkable areal increase with more than threefold (368.83%) change; mainly, the rapid expansion pattern was observed between 1994 and 2007 (Table 6 and Figure 6). The expansion of plantation forest was particularly dramatic in the 1994–2007 period (Table 6 and Figure 6).

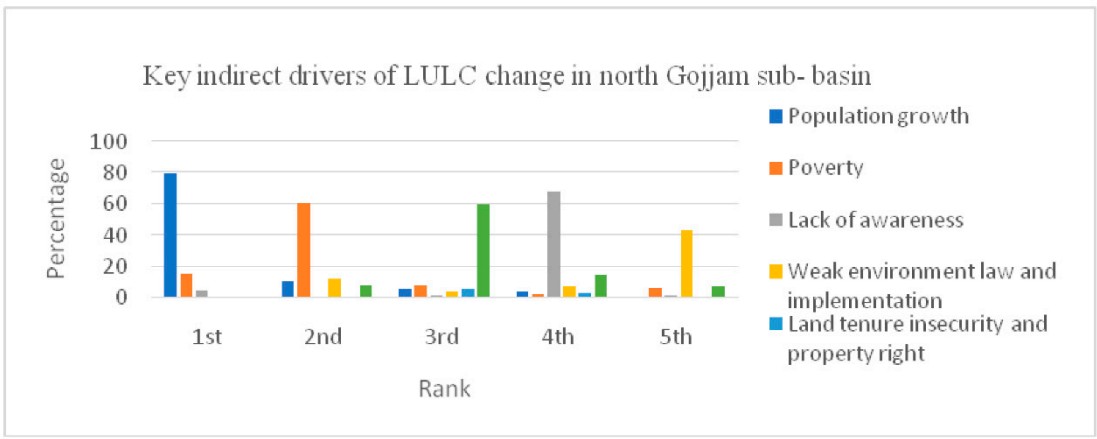

**Figure 6.** Indirect drivers of LULC change in the North Gojjam sub-basin.

However, other LULC types declined in areal size in the reference period. The areal size of natural vegetation and grazing land substantially declined from their original sizes (forest from 2.9 to 2% and shrubland cover from 13.4 to 8.2%; grazing land from 21.4 to 12.5%; Table 5). In terms of total area, the greatest land cover type loss was for grazing land (from 21.4 to 12.5%), followed by bush and shrubland (13.4 to 8.2%) and forest (2.9 to 2.0%) (Table 5). Local respondents also confirmed that grassland cover has consistently declined for the last three decades in favor of agriculture and buildings. According to them, communal grazing land cover has been converted to cropland, institutions (offices, schools, churches, and health centers) and rural towns. The decline in barren land was attributed to rehabilitation measures. Even though ponds in the sub-basin have been built, water bodies have experienced a reduction over the last 31 years. Local people also confirmed that water bodies have been declining in number and noted the disappearance and shrinkage of springs, streams, and swamps in the sub-basin. This could be due to a reduction in hydrological base flow associated with reductions in natural forest cover in favor of agricultural lands and with increases in water-hungry eucalyptus plantations.

**Table 6.** Magnitude and trend of LULC in the North Gojjam sub-basin.

| LULC | Magnitude of LULC Change in a Hectare | | | | LULC Change (Trend) in % | | | |
|------|-----------|-----------|-----------|-----------|-----------|-----------|-----------|-----------|
|      | 1986–1994 | 1994–2007 | 2007–2017 | 1986–2017 | 1986–1994 | 1994–2007 | 2007–2017 | 1986–2017 |
| CL   | 49,954.46 | 68,991.55 | 59,258.30 | 178,204.32 | 5.99 | 14.27 | 6.22 | 21.38 |
| WB   | 143.14 | −286.27 | −429.41 | −572.54 | 12.50 | −12.50 | −42.86 | −50.00 |
| BL   | −7299.94 | −14,456.74 | 18,178.27 | −3578.40 | −16.14 | −48.10 | 77.44 | −7.91 |
| GR   | −15,744.96 | −75,003.26 | −37,215.36 | −127,963.58 | −5.13 | −29.58 | −17.23 | −41.72 |
| FL   | −11,450.88 | 2147.04 | −2719.58 | −12,023.42 | −27.49 | 7.11 | −8.41 | −28.87 |
| PL   | 11,594.02 | 25,191.94 | 3864.67 | 40,650.62 | 105.19 | 333.77 | 8.08 | 368.83 |
| BSL  | −27,052.70 | −6727.39 | −40,936.90 | −74,716.99 | −14.07 | −17.57 | −25.84 | −38.87 |

Note: CL = cultivated land and settlement, WB = waterbody, FL = forest land, BSL = bush and shrub land, GL = grazing land, PL = plantation, BL = bare land.

### 3.2.3. Gain, Loss, and Persistence in Quantity of LULC Change

In the last 31 years (1986–2017), almost all LULC classes experienced some degree of change in the North Gojjam sub-basin (Table 7). In the entire study period, about 50.7% of the area that was covered with agriculture land in 1986 persisted in 2017 followed by grazing land (6.8%), bush and shrubland (5.5%), natural forest (1.2%), bare land (0.8%), plantation forest (0.4%) and water bodies (0.02%) (Table 7). The majority of agriculture land that was lost was converted into grazing land and plantations. Conversely, much of the land added to crop land was from grassland, bush and shrubland, and barren land. The largest gross increase in agriculture land (17.2%) was observed between 1994 and 2007 with the establishment of the Ethiopian People's Democratic Revolutionary Front

(EPDRF), followed by grazing land (5.6%) during the period of 1986–1994 (Table 8). The highest gross loss was observed for grazing land (12.8%) during the period of from 1994 to 2007, followed by agriculture (12.4%) during the period of from 2007 to 2017 (Table S1). Generally, the results show that the gain in the area of agriculture was nearly three times greater than the loss and much of the expansion came through encroachment on grazing lands. Land that was previously crop agriculture land most frequently changed to grazing land and plantation forests.

In the study period, about 3.6% plantation cover was gained from other LULCs, mainly from cropland and bush and shrubland (Table 7). This shows that most plantation expansion took place on cropland over the last three decades, but that some also encroached on lands previously covered with natural vegetation. Overall, land cover persistence was 65.1%, 63.8%, and 66.9% of the landscape for the periods 1986–1994, 1994–2007 and 2007–2017, respectively (Table S1). Over the entire study period (1986–2017), nearly 65.4% of the landscape did not experience land use/cover change, and 34.6% of the area showed a transition from one LULC class to another (Table 7). These findings suggest that during the study period, agriculture land cover was the most dominant in terms of persistence, followed by grassland land cover. However, this dominance may be attributable to the fact that agriculture land is represented by the highest proportion in the studied landscape. Similarly, the highest gain in the landscape was occupied by agriculture land followed by grassland, but the highest loss was under grazing land cover followed by cultivated land when compared to other LULC classes in the landscape.

**Table 7.** Land use/land cover transition matrices of the North Gojjam sub-basin from 1986 to 2017.

| LULC | CS | WB | BR | GL | FL | PL | BSL | Total | Loss | $T_c$ | $N_c$ | S | $G_p$ | $L_p$ |
|------|------|------|------|------|------|------|------|------|------|------|------|------|------|------|
| CL | 50.69 | 0.01 | 1.12 | 3.23 | 0.17 | 1.64 | 1.36 | 58.22 | 7.53 | 27.51 | 12.45 | 15.06 | 0.39 | 0.15 |
| WB | 0.02 | 0.02 | 0.01 | 0.00 | 0.02 | 0.00 | 0.01 | 0.08 | 0.06 | 0.08 | 0.04 | 0.04 | 1.00 | 3.00 |
| BL | 2.11 | 0.01 | 0.81 | 0.03 | 0.01 | 0.00 | 0.19 | 3.16 | 2.35 | 4.43 | 0.27 | 4.16 | 2.57 | 2.90 |
| GL | 12.67 | 0.00 | 0.83 | 6.87 | 0.03 | 0.60 | 0.42 | 21.43 | 14.56 | 20.18 | 8.94 | 11.24 | 0.82 | 2.12 |
| FL | 0.41 | 0.00 | 0.00 | 0.38 | 1.16 | 0.40 | 0.56 | 2.91 | 1.75 | 2.67 | 0.83 | 1.84 | 0.79 | 1.51 |
| PL | 0.30 | 0.00 | 0.00 | 0.02 | 0.05 | 0.35 | 0.15 | 0.77 | 0.42 | 3.69 | 2.85 | 0.84 | 9.34 | 1.20 |
| BSL | 4.47 | 0.00 | 0.12 | 1.96 | 0.64 | 0.73 | 5.53 | 13.43 | 7.90 | 10.59 | 5.21 | 5.38 | 0.49 | 1.43 |
| Total | 70.67 | 0.04 | 2.89 | 12.49 | 2.08 | 3.62 | 8.22 | 100.00 | 34.57 | 34.57 | 15.30 | 19.27 | | |
| Gain | 19.98 | 0.02 | 2.08 | 5.62 | 0.92 | 3.27 | 2.69 | 34.58 | | | | | | |

Note: CL = cultivated and settlement, WB = waterbody, FL = forest cover, BSL = bush and shrub land, GL = grazing land, PL = plantation, BL = bare land, $T_c$ = total change, $N_c$ = net loss, S = swap, $L_p$ = loss, $G_p$ = gain .

### 3.2.4. Net Change and Swap of LULC

Swap, net change, and total land use/cover change were calculated based on the change matrix result from the period of from 1986 to 2017 (Table 7). Thus, change as a result of consideration simultaneous gain and loss in different locations (swap) was highest for agricultural land (16%), followed by grazing land (11.3%) in the sub-basin. Likewise, change related to the quantity of net change was highest for agriculture land (12.5%) followed by grassland (9%) and bush and shrubland (5.2%) in the basin sub-basin. Similarly, the change attributed to swap and net change accounted for about 19.3% and 15.3%, respectively, for the last 31 years (Table 7). Moreover, the highest total change was observed on agriculture land (27.5%) followed by grassland (20.2%) and bush and shrubland (11%), and the observed total change was about 34.6%, providing both swap change and net change, which are vital to recognize the total transitions within the landscape (Table 7). The results imply that all LULC classes of the landscape experienced both swap and net change in the sub-basin over the last 31 years. The change attributable to quantity (net change) was lower than the change as a result of shifting location (swap change) of the landscape. Most of the change associated with agricultural land expansion may be from the largest class in the landscape. The findings of this paper are consistent with those of Teferi et al. [11] and Zewdie and Csaplovics [34].

### 3.2.5. LULC Change Distribution across Altitude

In the North Gojjam sub-basin, there was a distinct relationship between LULC distribution and altitude difference. In both study periods (1986 and 2017), agriculture was the most dominant land use/cover type within the 1500–3500 m elevation range. This shows that the proportion of agriculture was higher in the middle altitude, likely reflecting the suitability of these lands for crop production and settlement. The size of agricultural land cover increased in all altitude ranges for the period of from 1986 to 2017 in the sub-basin; however, the conversion rate was higher above 3500 m elevation (Table 8). Between 1500 and 3500 m, elevation grazing land cover was the second dominant land cover type after agriculture. During study period, the proportion of grazing land cover decreased up to 3500 m altitude, but it increased above this elevation (Table 8). The highest grassland cover conversion was observed between 2500 and 3000 m elevation in the two study periods (Table 8). Generally, the results show that grazing land areas are mostly located in the lower and upper parts of the sub-basin, which is due to the fact that the regions are quite rugged and not suitable for crop production and settlements.

The distribution of plantation forest was more concentrated in the upper parts of the sub-basin, above 3000 m altitude. Over the last 31 years, plantation forest cover has shown an increasing trend with different magnitudes across all elevation ranges below 3500 m, with particularly strong trends in the 2500–3500 m range (Table 8). Its cover expanded by more than six-fold during the reference years because of a decline in crop productivity in these areas. The proportion of bush and shrub land cover showed decreasing trends in all elevation ranges in the reference study periods, except for the 1500–2000 m elevation range. These decreases are most commonly associated with agriculture expansion in the middle of the sub-basin and overgrazing in the upper parts. The highest change to other land use was observed in the altitude range of from 3000 to 3500 m (Table 8). In part, this could be related to the fact that at lower elevations, the ratio of bush and shrubland cover was higher as compared to natural and plantation forest covers due to climate limitations on forest growth and the distance of some of these lands from settlements. Similarly, the size of bare land is greatest in the low land area (below 1500 m elevation) due to high soil erosion and shallow soil characteristics in this area. The bare land class decreased with increasing elevation in the sub-basin (Table 8). Over the last 31 years, waterbodies recorded the lowest areal extent in all elevation ranges and showed a decreasing trend across the elevations during the study period. This result is similar to the finding of Birhaneet al. [64], who found that the proportion of forest cover is lower than bush and shrub land covers in lower altitudes in Hugumburda national forest, northern Ethiopia. However, our result is similar to other findings of Birhanu et al. [65], who reported that farmlands were mostly located in the study area with 2000–2500 m of elevation in Ethiopian highlands.

### 3.2.6. LULC Distributions and Changes along Slopes

The spatial distribution and areal changes of LULC have a strong association with the slope in the North Gojjam sub-basin. More or less all of the LULC types were found across all six slope classes with different proportion in the years 1986 and 2017. Agriculture was the dominant cover type followed by grazing land in all slope classes during the study period. However, the highest cultivated land ratio was observed on very gentle (0–5%) and gentle (5–10%) sloped landscapes. Agriculture increased with a decreasing rate across the slope gradient over the last three decades. Comparably, increase in agriculture was mainly observed in the 0–5% and 5–10% slope classes (Table 8). The results show that expansion of agriculture covers is relatively higher in gentle and moderate slopes than in very steep slope gradients in the sub-basin. On the other hand, bare land decreased in almost all slope classes, but not above the 50% slope class gradient, where higher expansion of farm land was observed. This may be due to high soil erosion in very sloped areas. In contrast, grazing land area generally decreased in all slope classes. Conversion of grazing land to others was higher in very gentle (0–5%) and gentle (5–10%) slope classes due to it being occupied by cropland and settlements (Table 8). In the course of the study period, grazing

land decreased in all slope classes. Comparably, a decrease was mainly observed in 0–5% and 5–10% slope gradients (Table 8).

Planation forests were more commonly located in higher slope classes, particularly above the 20% slope gradient. During the last 31 years, while plantation forests increased in number, mainly above the regions of the 20% slope gradient, the rate of increase was more than four-fold in this elevation range. This implies that plantation forests consistently increased with the increasing rate of the slope gradient over the last 31 years. This shows that communities in the study area generally plant trees on steep slopes due soil erosion and low productivity of the land. Similarly, natural forest cover was mainly located at quite rugged landscapes and above 50% slope values, which are not suitable for cultivation. However, the highest regeneration of forest was common in the steep slope gradient (0–20% value) areas, while conversion to other land uses was higher in the steep areas (20–100% value) during the study period (Table 8). In general, the result shows that dense forest area was found on steep and very steep slope land, which is an unsuitable area for settlement, agricultural activity and grazing. On the other hand, higher concentrations of bush and shrub lands were located in higher slope gradient areas, mainly in very extreme steep slope areas in both study periods. Moreover, bush and shrub land cover decreased at the 50–100% slope gradients, but a higher conversion was observed at the 20–30% slope gradient (Table 8) due to farmland expansion. In general, in the sub-basin, agriculture and grazing land patterns showed fluctuating trends in all slope classes, but all vegetation covers (i.e., planation, forest, bush and shrubland) showed an increasing pattern with an increasing slope gradient, while agriculture, waterbody and bare land covers declined. The lowest area in all slope ranges was recorded for the size of waterbodies. Similarly, a consistent increase rate of change was observed for plantation forests, while a consistent decrement rate was observed for grazing land cover across all slope gradients between 1986 and 2017.

This study asserts that LULC change was significantly determined by slope inclination differences in the study area. The results of this study are in line with the study conducted by Birhane et al. [64], who reported that slope inclination is a major driving forces of LULC changes in Hugumburda national forest, northern Ethiopian highlands. A similar result was also reported by Zeleke and Hurni [60], in which cultivation land expanded to marginal areas as steep as >30% in the northwestern Ethiopian Highlands. The increase in forests can be observed to occur in steep slopes more than in other land uses, which may be due to the fact that these areas are inaccessible to human activities, unlike the moderate and gentle slope gradients. In contrast, Kindu et al. [66] noted that forests declined by half from about 63% in 1973 to 32% in 2012 on steep slopes in Munessa-Shashemene, Ethiopian highlands.

**Table 8.** Percentage changes in LULC along an altitudinal range and slope between 1986 and 2017.

| LULC Class | Altitudinal Range (m) | | | | | | Slope (%) | | | | | |
|---|---|---|---|---|---|---|---|---|---|---|---|---|
| | <1500 | 1500–2000 | 2000–2500 | 2500–3000 | 3000–3500 | >3500 | 0–5 | 5–10 | 10–20 | 20–30 | 30–50 | >50 |
| CL | 8.03 | 15.07 | 16.20 | 30.90 | 32.72 | 124.9 | 25.7 | 24.99 | 22.73 | 19.00 | 14.63 | 7.89 |
| WB | −85.5 | −57.7 | −17.0 | 34.42 | −82.35 | 0.00 | −52.8 | −46.3 | −49.5 | −52.0 | −69.9 | −92.4 |
| BL | 63.14 | −15.1 | −45.1 | −9.34 | 4612.0 | 22.52 | −38.6 | −33.8 | −17.6 | −3.23 | −4.00 | 3.17 |
| GL | −1.83 | −32.8 | −50.6 | −74.1 | −55.23 | 14.40 | −57.0 | −56.9 | −52.9 | −41.0 | −30.4 | −24.4 |
| FL | 30.15 | −6.99 | 0.03 | 11.42 | −58.68 | −92.2 | 329 | 1.03 | 21.51 | −38.1 | −20.7 | −26.7 |
| PL | 56.53 | 121.4 | 155.6 | 552.6 | 670.28 | 273.2 | 49.1 | 97.38 | 253.1 | 465.0 | 499.3 | 558.7 |
| BSL | −31.3 | 5.88 | −34.1 | −75.1 | −89.30 | −99.8 | −67.7 | −66.6 | −4.74 | −45.7 | −22.3 | 5.14 |

Note: CL = cultivated and settlement, WB = waterbody, FL = forest land, BSL = bush and shrub land, GL = grazing land, PL = plantation, BL = bare land.

### 3.3. Indirect Driving Forces of LULC Change

In addition to providing insights on LULC change patterns and their proximal drivers, engagement with local land users allowed us to place observed LULC change in the context of perceived driving forces related to demographics, policy, and other underlying social change. Land users in North Gojjam identified and ranked the direct factors of LULC

change. According to FGDs and key informant interview participants, the population in the North Gojjam sub-basin has increased with time. Reflecting this trend, about 79.5% of FDG participants ranked population increasein first place, as it was the root cause of LULC change (Figure 6). Local farmers in the informal interview reported that population is growing alarmingly due to a natural increase, and this results in continuous scarcity and fragmentation of cropland and grazing land uses in the sub-basin. Similarly, in the study area, poverty was ranked as the second most influential factor of land use/cover change by the majority (60.6%) (Figure 6) of FGD respondents. According to the local point of view, in the North Gojjam sub-basin, the majority of the population depends on unreliable subsistence crop–livestock farming systems and biomass energy for living. This is because there is no rural electric power, and other renewable energy sources (such as solar power or biogas) are also scarce, even in the towns. On the other hand, the livelihood of many poor people depends on the sale of firewood, dung cake and charcoal. According to them, in recent years, firewood and charcoal have become the most commercialized energy sources for both the rural and urban poor. Many people use forest resources in order to support their income and to repay loans from Amhara Microfinance.

Farmers in the in-depth interview also reported that lack of awareness and attitude about environmental management is another root cause of land resources change. According to the discussion results, the majority of local people were not aware of the negative impacts of forest degradation but seek livelihood benefits through the expansion of grazing and agricultural land. Still, most farmers in the study area considered resources (forest and grazing) for the government. Low implementation of environmental rules and extension service are other factors of ecosystem service change. Reflecting this, about 59.8% of FGD respondents ranked this in third place, as it is an underlying cause for direct drivers of LULC change in their locality (Figure 6). Weak law enforcement is another significant indirect driver of LULC change in the study area, and it ranked by 43.3% of FGD participants in fifth place as driver of change (Figure 6).

According to local viewpoints, infrastructure expansions such as schools, health centers, churches, roads, and offices have caused the conversion of communal grazing land into settlements over the last few decades in the study area. Moreover, one interviewee argued that gravel road accessibility and better market opportunities for firewood, charcoal, timber and timber products, and agricultural outputs facilitate land use/cover change. Furthermore, as observed during the field survey and confirmed in the interview, eucalyptus forest expansion is highly dependent on road and market accessibility in the study area. On the other hand, almost all farmers did not believe land tenure insecurity is a problem for the management of private cultivated land since they received land certificate care in the North Gojjam sub-basin; however, the problem of land insecurity has continued on communal (grazing and woodlands) lands.

## 4. Discussion

### 4.1. LULC Changes

LULC change analysis of the North Gojjam sub-basin for the last 31 years (1986–2017) shows a dynamic landscape change (Tables 5 and 6). The results show that removal of natural vegetation and cropland, settlement expansion, poor grazing, misuse of land management technologies, using biomass for fuel, topography and erratic rainfall were all identified as challenges related to LULC change in the North Gojjam sub-basin. Deforestation is among the key drivers of natural vegetation degradation in the sub-basin, particularly in the middle parts of the sub-basin because of the expansion of agriculture land to hillside-like areas in Ethiopia [67]. Agriculture and grazing land constituted the main land cover types, though the former has been by far the most dominant cover from 1986 to 2019. Of the total area of the sub-basin, agriculture shared about 58.2% in 1986 and 70.7% in 2017. Similarly, plantation forest cover increased more than threefold from its original amount, though eucalyptus plantations were the dominant type. In contrast, other LULC types showed a decreasing trend during this period: water bodies, bare land,

grazing land, forest land, and bush and shrubland. Dense forest has diminished and is currently observed in churches, riverbanks, and hillside areas. Studies conducted in different parts of the upper Blue Nile of Ethiopia also showed consistent results with this study—for example, Gebrehiwot et al. [12] in Birr and the Upper-Didesa watershed, Gashaw et al. [68] in the Andassa watershed, and Dibaba et al. [69] in Finchacachement. Additionally, Birhanu et al. [67] found an increase in farmland when compared with other land use types in the Ethiopian highlands. However, in contrast to our findings, Bewket [6] observed a rapid increase in forest cover in the Chemoga Watershed. The results also show that agriculture cover has experienced the highest gain, but grazing land showed the highest loss. Most stability and persistency were observed for agriculture cover. About 34.6% of the landscape experienced transition from one class to another LULC during the 31 years. Of this, about 15.3% is attributable to net change, while 19.3% is due to swap change. This implies that the wide range of LULC conversions occurred on agriculture land and grazing land use types. Similarly, Teferi et al. [11] reported that the highest gain of agriculture land was obtained from grazing and shrubland from 1972 to 2009 in the Jedeb watershed.

Generally, the results imply that agricultural expansion increased crop production by increasing farm size, but it is also a result of deforestation and high soil erosion. Similarly, firewood collection and charcoal production are among the sources of livelihood for the rural poor, particularly youths, but they are also a cause of forest clearances.Eucalyptus planation is one of the key sources of income for rural smallholders in the study area, but it has negative ecological impacts on water and indigenous trees. The decrease in percentage change of natural forest land stopped in 2007 due to the absence of suitable natural forest land for further conversion to agriculture land and settlement land and the implementation of local rules and regulations with respect to forest management. Land resource changes are driven by several root causes in the North Gojjam sub-basin. The most cited root cause drivers to LULC changes are associated with rapid population growth, persistent poverty, and poor governance.

### 4.2. Local Perceptions Compared to Satellite Mapping

Overall, the qualitative land cover change narratives reported by locals and the quantitative remote sensing approach yield generally similar results. This enhances our confidence in both methods. Given the similarity of the results, the advantages of using the remote sensing method are its spatial completeness, quantitative nature, and objectivity. However, this approach includes some sources of error and gives no information on why changes have occurred [70]. In contrast, the reasons behind LULC changes were explained in depth by local land users, and the types of the drivers that are more likely to cause change were also identified via interaction with the local land users. For example, local land users provided an in-depth explanation as to why grazing land changed to cultivated land and why dense forests are located at riverbanks and hill sides in the sub-basin. This implies that the limited remote sensing quantitative data can be triangulated by qualitative data from local land users' knowledge. In this study, the advantages of multitemporal remote sensing data over local perception were the ability to quantify the extent, rate, and spatiotemporal pattern of LULC change numerically. However, the local viewpoint method permitted the in-depth discussion of why these processes happened during the specified study period. Therefore, the advantages of local knowledge allow for the identification of the key drivers of LULC change, mainly indirect drivers which cannot be easily determined by geospatial analysis. Using local viewpoints, the key drivers of LULC were ranked from the most to least observed in the study area. Geospatial methods assist the analysis of LULC dynamics in less time and with low costs in the entire sub-basin. However, local perception is time-consuming and limited to specific local area that farmers become aware of over their lifetime. This implies that studies from local interviews are mainly locally specific, which can cause bias in analysis of a large study area over long time intervals.

### 4.3. Indirect Driving Forces of LULC Change

Proximal driving forces for LULC change in Ethiopia are varied and context-specific. Such driving forces concern both human and natural drivers [10]. However, rapid population growth is significantly related to increasing demand of scarce land resources [10,58,71]. The population in the North Gojjam sub-basin has increased. Likewise, the official district population data for the sub-basin and surrounding villages show a population increase of from 2,705,913 in 1994 to 3,565,892 in 2016 [9,72]. Reflecting this trend, FDG participants and key informants in interviews confirmed population increase as the main root cause of natural resource changes in the North Gojjam sub-basin (Figure 6). As result of continuous population growth, farm land owned by the parents is continuously divided by the number of descendants before or after their death, and, thus, land fragmentation increases, but its size decreases. This has resulted in resources scarcity and has furthered natural resource degradation in the study area. Population growth results in cultivated land encroaching at the expense of forest land, rangeland, bush, and shrubland in the sub-basin. Natural forests have also been lost due to population-associated increased demands for firewood, building materials, and other household furniture and farm equipment. Government redistribution of communal grasslands to landless youths is the major cause of the contraction of grassland to agriculture land. Our results are in agreement with those of recent studies focusing on other parts of Ethiopia, which have indicated that population growth is the most influential factor in LULC change [6,7,46,72]; however, this conclusion is not similar to the findings of Shimeles [17].

In Ethiopia, the majority of rural communities are poor and depend on mixed subsistence crop–livestock farming systems, biomass fuel energy, and the inability to invest in livelihood diversification strategies [10,69]. Similarly, poverty is the second root cause of LULC change in the North Gojjam sub-basin (Figure 6). The majority of the population in the sub-basin depends on mixed subsistence crop–livestock farming systems supplement by the sale of forest products. Many of the inhabitants of this area borrow money from Amhara Microfinance (ACSI), and they have no alternative household energy sources to substitute biomass fuel. The livelihoods of many poor people depend on selling of firewood, dung cake, and charcoal. The majority of local people were not aware of the impact of natural resource degradation. As a result of this, there is high vegetation cover conversion in the sub-basin, while there is a significant scarcity of natural vegetation cover. These results are similar to those of the study conducted in the Bale Mountain by Hailemariam et al. [36] and in the Gog district, Gambella region by Oo et al. [67].

Low implementation of environmental rules and extension services are other factors of ecosystem service change in the North Gojjam sub-basin. Evidence from local respondents shows that there was considerable weakness in implementing environmental law and extension services at ground level in the sub-basin. Agricultural activity still depends on rain feed as well as low input and output due to lack of improved seed varieties and fertilizer accessibility and water for irrigation. Improved livestock herds and feeding systems also remain of poor quality. Moreover, there are no well-organized off-farm activities, such as beekeeping or poultry farming, to integrate landless youths and to decrease population pressure on land by extension service staff. Similarly, about 59.8% of FGD respondents ranked this issue in third place as it is an underlying cause of LULC change in their locality (Figure 6). Weak law enforcement, on the other hand, was a significant indirect driver of LULC change in the study area and was ranked by 43.3% of FGD participants in fifth place. From the discussion, it was observed that key indicators, such as illegal logging, charcoal making, free grazing, and firewood collection from the natural forest were not implemented, as there were restrictions, rules, and regulations in place in the sample districts.

Local land users confirmed that infrastructure expansions such as schools, health centers, churches, roads, and offices caused the conversion of communal grazing land into settlements over the last few decades in the study area. Moreover, one interviewee argued that gravel road accessibility and better market opportunities for firewood, charcoal,

timber and timber products, and agricultural outputs facilitate land cover change study area. Furthermore, as observed during the field survey and confirmed in the interview, eucalyptus forest expansion is highly dependent on road and market accessibility in the study area. Many changes have been observed in land ownership in Ethiopia, which continues to provide uncertainty to land users. The traditional feudal system was followed by a communal form of government ownership, and polices regarding the security of tenure and land resource rights remain to cause confusion at the regional level [10]. According to Berry et al. [10], land tenure arrangement influences Ethiopian farmers' decisions concerning land management in relation to whether the land is managed privately or communally; the ability to manage the land; and the ability to transfer land by sale, lease, or bequest. Similarly, key informants highlighted that, historically, land tenure insecurity was one of the drivers of land degradation in the study area. However, in current time, land tenure insecurity is not much of a problem with regard to land management, except communal (grazing and woodlands) lands, as land certification cards can be obtained.

The findings of this study suggest that unless driving forces and pressures are ceased, LULC change in this sub-basin will continue and further contribute to land degradation. This could be a major threat to agricultural sector development and also a potential threat to environmental sustainability and the balance of ecosystems in general, as well as the biodiversity of the study area in particular. Moreover, the fact that the study area is located in the transboundary upper Blue Nile basin emphasizes the fact that the prevailing LULC change and associated problems observed in the area have environmental implications for other communities and at the regional, national, and international levels. Finally, this study has highlighted that integrated use of remote sensing and GIS technologies with land users is important for the triangulation of their relationship, and it has emphasized the importance of increasing the robustness of quantitative information regarding remote sensing with quantitative local viewpoints in order to improve our understanding of the process of land use/cover change.

**Supplementary Materials:** The following are available online at https://www.mdpi.com/2073-445X/10/1/68/s1: Table S1: Land use/land cover transition matrices of north Gojjam Basin.

**Author Contributions:** All authors have made substantial contributions equally to the design of the research work and the acquisition, analysis, or interpretation of data. All authors have read and agreed to the published version of the manuscript.

**Funding:** This research was funded by NILE-NEXUS: Opportunities for a sustainable food–energy–water future in the Blue Nile Mountains of Ethiopia. Belmont Forum. Addis Ababa University and John Hopkins University have also supported this research.

**Institutional Review Board Statement:** Informed consent was obtained from all subjects involved in the study.

**Informed Consent Statement:** Not applicable.

**Conflicts of Interest:** The authors declare no conflict of interest.

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
