# Peer review of "Land Cover Change in the Blue Nile River Headwaters: Farmers’ Perceptions, Pressures, and Satellite-Based Mapping"

_land, doi:10.3390/land10010068_

Round 1

Reviewer 1 Report

The manuscript entitled "Land cover change in the Blue Nile River Headwaters: Farmers’ Perceptions, Pressures, and Satellite-based Mapping", presents an interesting work. In general, the manuscript should be considered in major revision, but resubmit are needed to make the article more reasonable.

  1. Correct the format of the manuscript in the text according to the Journal’s format.
  2. Please check Figure 2. Some words are not clear.
  3. Please add the land use/cover change maps and explain the distribution of forest change from 1986 to 2017.
  4. This study used too many tables to show land use/cover composition. The statistic number only can provide optimal results, but in this research, we should get more information from spatial results. Please read the relevant papers and change them.
  5. First, the writing needs to be improved. Although the English level is not bad, there are still a lot of sentences with a weird syntax.
  6. Reference is too strange and very terrible, where is reference 2, 3, 5….? Please revise them based on the Journal’s format.
  7. Line 36, km2….
  8. Line 258, reference order
  9. In Table 2, please keep the same order of All land use types with land use/cover maps.
  10. The manuscript is poorly organized, the text overlapping images, weird alignment, etc. It’s hard to see any scheme in this piece of writing and very confused. Please consider rewrite it.

Date of manuscript submission

18 November 2020

Date of this review

28 November 2020

Author Response

Dear Sir,

We have tried to correct all the suggestions needed. We have attached the responses for your information and corrected the article accordingly.

Regards,

Belay Simane

Reviewer 2 Report

Review on manuscript:

Ewunetu, A., Simane, B., Teferi, E. and Zaitchik, B.F. Land cover change in the Blue Nile River Headwaters: Farmers’ Perceptions, Pressures, and Satellite-based Mapping.  Land 2020.

Summary

This manuscript focuses on land use/land cover change in the North Gojjam sub-basin, in the Blue Nile River Headwaters, Ethiopia. The authors use satellite imaging, remote sensing and also interviews from the local people in order to evaluate the status of the land cover/land use change and to identify the key factors that drive these changes. Based on their findings there are many factors that cause, interact and eventually lead to these changes, such as deforestation, overgrazing, illegal logging for fuels or to increase income by selling, poor legal management and climate and environmental factors. Based on their findings they emphasize that the opinion, knowledge and conversation with the local people is an important tool - along with surveying and mapping data collection methods - in understanding the factors and causes that affect the assemblages of land use/land cover changes in an area.      

Merits

The study focuses in land use/land cover change that is an interesting and important issue of ecology, topology and agroecology, due to the increasing climate change and continuous degradation of the environment. The analyses that authors use are appropriate. The authors use two deferent approaches to examine the driving forces and factors that lead to land use/land cover change in an area· mapping techniques and interviews from locals in the study area. The study should separate Results from Discussion because that will make the manuscript more comprehensible and give the study a flow that the reader will find easier to follow and understand. There are important issues with the References section that need correction. Tables and Figures are also in need of modifications and little “beatification”.

The manuscript would benefit from English language editing.

Critique

Abstract

Abstract is a brief version of the manuscript and should provide sufficient information about the study and to entice readers to read further. Thus, the authors must consider that the first referral to LULC (Line 15) must be with the full term “Land Use/Land Cover”. The subject of the study will be clearer to the readers.

Line 17. The authors refer to “seven major LULC types”. The authors might find it be useful to add a short listing of these factors, as this will help the reader to create an idea about the subject.

Introduction

The Introduction provides a good background on the main subject. The objectives of the study are clearly defined in the last paragraph of the Introduction. The whole section needs some clarifications as of the causes of land use/land cover changes in previous studies and further addition of references and comparisons regarding them will be useful to the readers. The Introduction section would be benefit from English language editing.  

Line 37. The authors mention that “The majority of vegetation cover... and settlement covers”. The authors may wish to remove the word “covers”, as “settlements” describe sufficiently the use of the area.

Line 37. Citation [2] is absent from References section.

Line 41. Citation [3] is absent from References section.

Line 43. Citation [5] is absent from References section.

The authors are mentioning that “several studies have been conducted on the biophysical state of change in the basin using remote sensing and GIS technologies” (lines 51-57). However the authors just cite these references without presenting examples of their findings. Are these techniques able to detect changes in land use/land cover based on previous knowledge?

The authors also mention that “the findings were contradictory” (lines 55-57). The authors need to provide more information and details to support and explain this statement, as it will be useful to the reader. It will also be important later in Discussion as it will be helpful in comparison and evaluating the results of the current study.

On that note, in the same paragraph is later mentioned by the authors that “drivers of LULC change are complex” (lines 57-61). Can the authors provide examples of these drivers and their complexity?  “The causes ... are time and location specific”. Can the authors provide examples of the specific causes? It might worth presenting these examples because this will be of major importance when discussing the findings and results of this study in comparison and juxtaposition with previous studies.

Line 61. The authors mention: “If there are interventions by the actors”. I think that the authors meant to write “by the factors”.

The authors mention that “This is also true in the upper Nile basin of Ethiopia due to continuous human intervention” (lines 62-63). The authors may wish to provide references to support this claim.

The last lines of this paragraph (lines 81-84) describe the overall structure of the manuscript. In my opinion this part is not necessary and could be removed. Otherwise this paragraph will need modifications in order to clarify and specify the sections that follow.

Materials and Methods

Description of the Study area

The authors use traditional Ethiopian terms to describe the climate zones of the study area· Wurch, Dega etc (lines 91-95). Since the study aims to an international audience, it might useful and less confusing to use other terminology to explain the climate zones.

Line 96. The authors mention that “according to National Metrological Agency (2018) ...” This statement needs reference. The citation is not present in References.

Line 101. The authors mention that “according to CHIRPS data ...” This statement needs reference. The citation is not present in References.

Figure 1. The resolution and quality of the map is low. Toponyms and places that are mentioned on the map are difficult to read. This figure needs processing and improvement in order to be more appealing and easy to read.

Data Sources and Methods

This section is presented in paragraphs each one starting with a title in bold lettering. I suggest that the authors could modify it in order to build flow in the text and reading, and thus be more attractive to the reader, more like a continuous story and not divided by titles.

Community Survey (lines 122-134). In accordance with the authors this is one of the most important aspects of their study. They combined the knowledge, experience and opinion of the local people with data in order to have accurate conclusions concerning the land use/land cover change. The authors do not provide the place and time period during which the surveys were made. The authors mention nine villages, what are the names of these villages? Why did the authors choose these areas? What were the questions that guided the discussion within the group of subjects?

The authors also state that “On the other hand, in-depth interviews also held with farmers” (lines 135-137). How many farmers did they participate in these interviews? Are they part of the above mentioned group surveys? When and where these interviews took place?  Where representing all 9 villages where the survey took place? The authors need to provide to readers more information about this as is one of the main subjects of their study and the part of it that differentiates it from previous studies.

Satellite images (lines 143-149). The authors state that “Images were selected for years that align with major events that happened in the study area” and further discuss this subject in this paragraph. It would be better to move this description in Introduction section along with analysis and information regarding the selected periods. Materials and Methods is not the place for such an analysis. This information will also be helpful later in Discussion.

Ground truth data (lines 153-154). The authors state that “Reference data for the 1986 and 1994 images was collected based on local elder knowledge”. Were the elders part of the Discussion groups that participate in the Community surveys? How this knowledge was collected, using interviews or literature? The authors must provide more information and support their statement using references.

The authors also mention that “image was collected from Google Earth Pro map” (line 155). The authors need to provide reference for this in the References section and information about the version of the software they used. This also applies in line 157 “interpretation of Google Earth Pro map”.

The authors mention that “2500 sample points were collected from the representative of LULC classes using a Garmin GPS instrument. Of this total sample points, 547, 587, 724 & 738 ground truth points were used for accuracy assessment of the year 1986, 1994, 2007 and 2017 images, respectively” (lines 158-161). The authors must provide how this selection was made, if it was random or if certain criteria were made. The authors also mention representatives of LULC classes, but the details for these classes are mentioned later in the manuscript. It would be easier for the reader to explain classes and their categorization in the Study area section in order to familiarize the readers with the terms in use. The authors must provide details about the Garmin GPS instrument that was used.

Satellite Image Pre-processing (lines 168-169). The authors state that “Image preprocessing can be done through various radiometric and geometric correction techniques”. This statement needs support with reference.

Line 173. The authors state that they acquired images from Earth Explorer Website. The authors must add a citation in References section.

Line 201. The authors state that they used QGIS software. The authors must cite QGIS Software in References section and provide details about the version of the software they used in their study.

Lines 207-208, 215, 240. The authors state that they used ERDAS IMAGINE. The authors must cite ERDAS IMAGINE Software in References section and provide details about the version of the software they used in their study.

Lines 225, 228, 235. The authors state that they used ArcGIS software. The authors must cite ArcGIS Software in References section and provide details about the version of the software they used in their study.

Line 227. The authors state that they acquired data about altitude and slope from ASTER GDEM. The authors must cite this in References section.

Satellite Image Classification (lines 237-272). I suggest that the “Satellite Image Classification” section could be moved after “Description of Data Area” section. In that way it gives the reader the opportunity to be familiarized with the LULC classes that are mentioned in the previous sections and it makes comprehension easier.

Analysis of LULC change detection (line 284). The authors state that “matrix methodology used by [9, 31]”. The authors must use in-text citation style and quote the author's last name and the year of publication, when they refer to a previous study inside the text of the manuscript. The same applies in line 290 (“according to [47]”) and to line 309 (“and recently applied by [9]”).

Figure 2. The overall quality of this flow chart is low. Terms are not easily read, fonts in text-boxes are not the same. This figure needs improvement in order to be more appealing and easy to read. The authors must also explain in Figure heading details about TM1986, TM1987, ETM+2007, OLI2017.

Results and Discussion

This section needs improvement. I would recommend dividing Results and Discussion into two separate sections. The authors may find this way easier to present their results and then in a different Discussion section to have freedom to analyze them in depth. The authors might find easier to correlate and compare their results in an autonomous Discussion section. The authors should describe in simple terms what the data show, because often the analysis is confusing and references to tables and figures are missing.

As I already mentioned authors use the traditional terms for climate zones in Ethiopia and this is confusing, since at all times they also use the corresponding common term in parentheses.  Regardless of this the section needs an extensive editing in English language.

Table 3. The heading of the Table is not representative of the table’s data. The authors must also include in heading details about the data of the table. The authors could replace “Respondents in percentage per rank”, with “Percentage of Respondents per class”.  Since, as I mentioned in previous paragraph, these results are as of major importance in this study they have to be properly presented. The purpose of the number rating above results (1-5) it is not easy to understand. I suggest that the authors could present these results in a Pie or a Column Chart. In Data section (lines 122-125) authors mentioned that 18 Group Discussion surveys were made in 9 villages and also some farmers’ opinion was included. How did people from different villages responded? That would be very interesting topic in Discussion. How did people from different social class responded? This would also be interesting. The authors have a very interesting data to analyze but are poorly presented.

Table 4. The table in LULC column contains twofold the term “GL, grazing lands”. “BR, bare lands” is missing.

Line 396. As I already discuss in previous sections the authors should replace “according to [55]” with the proper in-text citation.

Line 410. The authors have already use Figure 2 in previous figure heading. This figure should be renamed as “Figure 3”.

Line 424. As mentioned above and for the same reasons this should be renamed to “Figure 4”.

The authors mention that “During the same period, land covered by plantation and natural forest 431 increased by 333.8% and 7.1% change, respectively” (lines 431-432). It might be useful for the reader that the authors should specify the Table that contains this information, thus making it easier to follow the presentation of the results. The same applies to results presented in lines 439-441 and 451-453.

Lines 469 and 470. These Tables are named Table 5 and Table 6. However Table 5 refers to an above data table (line 426), so the authors must change the names of these two tables to Table 6 and Table 7 respectively. The authors also must include a note explaining the abbreviations that are been used in these tables. They must then also check thoroughly the text of the study and change the above as to be corrected. The authors must the check the rest of tables and figures in the manuscript and rename them correctly and accordingly to the right order.

Lines 466, 501, 558. As I already mentioned “reported by [6]”, “according to [9]” and “to the findings of [58]” should be changed to correct in-text citation.

Line 565, 566, 657. The numbering of the Figures needs calibration as I repeatedly mentioned. The heading of this figure is wrong, because it states that contains data from 2017 but the data are from 1986. It would be more appealing to the reader if the axes in diagrams and figures are the same. In this instance the “Area (%)” axes are different in these similar in data-type figures.

Line 615. The authors must include a note explaining the abbreviations that are been used in the table.

Lines 651-655. The authors need to rephrase this statement as to connect the observation with the result. Land fragmentation due to inheritance rights (and thus smaller land rights per owner) leads to land encroaching. It is a very interesting observation and the authors may wish to edit this conclusion as to be better understood.

Lines 667-559. The authors mention that “the majority of the population depends on mixing subsistence crop-livestock farming systems supplement via selling of forest products, and loans from Amhara microfinance (ACSI)”. I understand that the authors want to support their claim that people are using forest resources in order to support their income and maybe to repay their loans from Amhara Microfinance. The authors may wish to rephrase this statement because their conclusion it is not made clear.

The authors state that “However, currently, land tenure security is not much problem on land management, except communal (grazing and woodlands) lands due to land certification” (lines 711-714). This statement needs support with references.

Conclusions and Policy Implications

In my opinion contents of this section must be merged and further enriched with parts of the Results and Discussion section above. As I already mentioned Results must be a separate section, and Discussion section should be merged with Conclusions. This will help the authors to tell a coherent story without separating the factors, drivers and cases in different sections as these often correlate. Discussion should gather all the information into a single whole and authors should describe the overall story formed.

The manuscript would benefit from extensive language editing. The authors must also try to compare their findings by referencing published research as to confirm and compare their findings. There are several instances where assertions are made that are not substantiated with references. This will help the authors to evaluate the observed trend and explain the significance of their results.

References

This section needs thorough check both in editing and correction of the cited bibliography.

References [2], [3] and [5] are missing from References section.

Reference [8] is cited two times. The second must be removed.

Reference [14] and Reference [19] are the same (Sileshi et al. 2012). One must be removed and the reference in the main text should be fixed.

Reference [15] and Reference [26] are the same (Simane et al. 2016). One must be removed and the reference in the main text to be fixed.

Reference [21] and [46] are the same (Fairhead and Leach, 1995). One must be removed and the reference in the main text to be fixed.

Reference [56] and [63] are the same (Oo et al. 2017). Suggestion is same as above.

Useful References

Akinyemi, F.O. (2017) Land change in the central Albertine rift: Insights from analysis and mapping of land use-land cover change in north-western Rwanda. Applied Geography 87: 127-138.

Birhanu, L., Hailu, B.T., Bekele, T. and Demissew, S. (2019) Land Use/land cover change along elevation and slope gradient in highlands of Ethiopia. Society and Environment 16, 100260.

Demissie, F., Yeshitila, K., Kindu. M. and Schneider, T. (2017) Land use/Land cover changes and their causes in Libokemken District of South Gonder, Ethiopia. Society and Environment 8: 224-230.

Hailu, B.T., Maeda, E.E., Heiskanen, J. and Pellikka, P. (2015) Reconstructing pre-agricultural expansion vegetation cover of Ethiopia.  Applied Geography 62: 357-365.

Kindu, M., Schneider, T., Taketay, D. and Knoke, T. (2013) Land Use/ Land Cover Change Analysis Using Object-Based Classification Approach in Munessa-Shashemene Landscape of the Ethiopian Highlands. Remote Sensing 5: 2411-2435.

Tolesa, T., Senbeta, F. and Kidane, M. (2017) The impact of land use/land cover on ecosystem services in the central highlands of Ethiopia. Ecosystem Services 23: 47-54.

Author Response

Dear Sir,

Thank you so much for your professional inputs. We have addressed all the comments accordingly and corrected the article as appropriate.

Regards,

Reviewer 3 Report

The land cover change analysis is very crucial to maintain sustainability among natural resources and their vaue in the socio-economic domain. This study talks about land change analysis integrated with local people perceptions. The authors have done a good job in assesing the land cover chnage in Blue Nile River headwaters. There are few changes that need to be implemented before considering it for publication. I have two different kinds of suggestions: One is formatting based; and other is the reasoning required from the authors.

Format based changes:

Please check double spaces which are there in the manucript at many places. Example: Line 134, 257,344,357, etc. Check throughout the paper.

Please check spelling throughout the paper and providing full form of the abbrviations used. Examples: Line 97: varies not various; 101: Define CHIRPS; Line 128: design not deign; Line 135: define DA's; Line 201: TOA not ToA; Line 253: ease not easy: Line 343: drivers, not derivers, Table 5: Align the LULC types in teh first column.

The refernecing in the text is shown as power at almost all the places. Please correct the text to normal. Example: Lines 96,132,145,150,201, 199, 166, etc. There is no consistency in referencing in the text.

Please do not start a sentence using "via". Example: Lines 232,278.

Reason-based comments:

Line 35: Please provide a refernece for the loss in forest, woodland and grassland.

Line 137: There is no information about teh checklist used to guide the researcher for the farmer's interview. What type of questions were included. How does the quality of interviews impact on land cover change analysis. What if extending thsi method to other rgeions where a lower quality of perceptions would be received. How important is it to be involved in change analysis.

Line 158: Sample points: did you use a single pixel as sample? If yes, then it is  not recommended as standard sampling methods. I see that teh authors have mentioned polygons in line 247. What is the size of this polygon? How many pixels were used as a group or polygon to assess 30m maps?

Line 160: Different sample numbers are mentioned that were used in the validation process. How these numbers were derived? However, in Line 281, it has been mentioned that 30% samples were used as validation. These numbers does not either represent 30% of 2500 for all the years. Please do not use sample points and groung truth. You can use samples and ground reference data because they are never points ideally and not true to the ground also.

Line 261: What classifictaion scheme was used? Is it not a standard form of scheme that can be compared with other classified maps in the future studies.

Line 277: "A set of reference pixels".. Please describe the number of samples or pixels used in teh accuracy assessment.

Line 299 to 301: Please describe the equation more clearly. It is not clear from the text, why the equation for swap change was used. Please provide clear details.

Line 306: Flowchart: how ancillary data was used in accuracy assesment. it should be reference data i think.

Line 310: Why did you take log of amount of land cover at a particular time?

Line 23 and 565: Why plantation forest change was incresed with increasing altitude? With increasing altitude, how come it was easy to perform plantation in the sub-basin.

Author Response

Dear Sir,

Thank you so much for your very professional edits and comments. We have tried to address all the comments as appropriate and corrected the article.

Please find attache the detailed responses.

Regards,

Round 2

Reviewer 1 Report

You have significantly improved the paper.The manuscript is organized and written fairly well.

Author Response

Dear Editor of LAND,

Please find attached the response to your comments. We have made the necessary corrections to all your inquiries. We have also addressed all the suggested changes by the English editor. 

Respectfully,

Belay Simane
